# Collective patterns of social diffusion are shaped by individual inertia and trend-seeking

Mengbin Ye [1,2 ✉], Lorenzo Zino [2], Žan Mlakar [3], Jan Willem Bolderdijk [3], Hans Risselada [3], Bob M. Fennis [3] & Ming Cao [2 ✉]

Social conventions change when individuals collectively adopt an alternative over the status quo, in a process known as social diffusion. Our repeated trials of a multi-round experiment provided data that helped motivate the proposal of an agent-based model of social diffusion that incorporates inertia and trend-seeking, two behavioural mechanisms that are well documented in the social psychology literature. The former causes people to stick with their current decision, the latter creates sensitivity to population-level changes. We show that such inclusion resolves the contradictions of existing models, allowing to reproduce patterns of social diffusion which are consistent with our data and existing empirical observations at both the individual and population level. The model reveals how the emergent population-level diffusion pattern is critically shaped by the two individual-level mechanisms; trend-seeking guarantees the diffusion is explosive after the diffusion process takes off, but inertia can greatly delay the time to take-off.

[1] School of Electrical Engineering, Computing and Mathematical Sciences, Curtin University, Perth, Australia. [2] Faculty of Science and Engineering, University of Groningen, Groningen, Netherlands. [3] Faculty of Economics and Business, University of Groningen, Groningen, Netherlands. ✉email: mengbin.ye@curtin.edu.au; m.cao@rug.nl

Social conventions, such as the use of handshakes versus bowing for greetings[1], or accepted grammatical rules and word meanings in languages[2,3], are fundamental aspects of society and culture[4,5]. The value of a convention is tied to its widespread adoption and acceptance, because individuals mostly derive benefit from coordinating to select the same option, rather than because of an intrinsic advantage of a particular option[1,4–7]. The phenomenon of social diffusion is an important mechanism for the change and evolution of conventions, whereby an alternative is proposed by a minority and subsequently diffuses widely across the population to replace a status quo[3,8,9].

Individuals can play substantially different roles during social diffusion. In many instances, a new alternative to the status quo is stubbornly promoted by a committed minority[10–12]. Within the remaining uncommitted population, a fraction of individuals whom we term explorers may first test out the alternative. If the combination of committed minority and explorers adopting the alternative reach a critical mass[10], this may then spark a rapid diffusion to the rest of the population, whom we term non-explorers.

Aside from works focusing on empirical data and laboratory experiments[8,10,13–15], mathematical models have emerged as a valuable framework for studying social diffusion[3,16,17]. The first models, including the classical Bass model and its variants, were population models that focused on capturing the diffusion process at the macroscopic (societal) level[3,16,18–20]. A strength of such population models is their ability to use only a few parameters to predict the macroscopic features of diffusion, including the ubiquitous S-curve, which describes the typical adoption pattern in the population over time[16,21]. Although population models can successfully describe how diffusion occurs, they are limited in their ability to explore why diffusion occurs.

Agent-based models (ABMs) have been proposed as a valuable paradigm to address this limitation, and are becoming increasingly popular[22–26]. ABMs consider a population of agents with specified dynamics, and have several advantages over population models. One key advantage is the ability to hypothesise and test how microscopic dynamics at the agent (individual) level may lead to the emergence of complex macroscopic phenomena at the population level as agents interact over time[24]. Literature on complex contagion[8,27–30] has underlined the need to focus on individual-level dynamics, which can be crucial during social diffusion processes. Another key strength is the ability to directly study individual-level factors known to significantly impact the diffusion process, such as population heterogeneity[31], the structure of social contacts[8,32,33] and targeted intervention strategies[25,34].

Social conventions typically afford individuals the ability to revise decisions on selecting different options, and each individual's decision can in turn affect others' decisions. For example, in the context of language conventions, an individual can decide between the status quo or alternative spelling of a word (e.g. 'centre' versus 'center'[3]) each time he or she uses it. Thus, human decision making frequently plays a central role during social diffusion, and game theory offers a robust framework for modelling human decision making. Consequently, ABMs utilising game theory have emerged as a powerful paradigm to study social diffusion[31,32,35–38]. These models are typically based on a social coordination mechanism promoting collective behaviour that captures the real-life desire to conform to others and reach a consensus on social conventions[17,39].

However, other important behavioural mechanisms besides social coordination can feature prominently in each individual during decision-making processes. The social psychology literature has pointed out the ubiquitous presence of two such behavioural mechanisms. The first is that individuals often prefer to stick to their current decision, which we term inertia (sometimes referred to as status quo bias[40]), and the second is that people tend to follow the trends observed in the population, which we term trend-seeking (recently discussed in some contexts as dynamic norms[41]). For each of the two mechanisms, there is empirical data supporting their presence during decision-making processes, and different theories have been put forward as to explain why such mechanisms are present[40–45].

Existing ABMs, including game-theoretic models, do not consider inertia and trend-seeking. In this paper, we show that as a consequence, such models cannot capture important macroscopic features of many real-world social diffusion patterns– such as a long delay before the diffusion process takes off followed by an explosive transition– unless unrealistic agent-level assumptions are made. In turn, these unrealistic assumptions produce individual-level decision-making patterns that are inconsistent with the aforementioned literature. The consequences of such inconsistency on the real-world application of existing ABMs are twofold. First, their calibration may be more challenging, since parametrisation cannot be driven by individual-level data. Second, it hampers their ability to explore factors, such as network structure effects and individual-level interventions.

In this work, we show that these issues can be resolved by introducing a game-theoretic model that, besides social coordination, explicitly incorporates the two behavioural mechanisms of inertia and trend-seeking. To begin, we conducted a multi-round group experiment with 20 trials using a setup inspired by Centola et al.[10]. The experimental data provided further statistically significant motivation, building on the social psychology literature discussed above, to include inertia and trend-seeking mechanisms in our ABM. Moreover, our experimental data was used to parametrise our model. The data further identified heterogeneity in how participants were affected by the two behavioural mechanisms. Some participants, termed explorers, were less affected by inertia and more susceptible to trends than others, termed non-explorers. After using our experimental data to calibrate the model, we employed numerical simulations to examine how individual inertia and trend-seeking shape collective patterns of social diffusion at the population level, leading to change and evolution in social conventions. We found that inertia produces a delay in the time taken for the diffusion process to take off. Second, and surprisingly, the presence of trend-seeking results in diffusion that is always explosive: once the diffusion process takes off, the alternative spreads rapidly irrespective of the initial delay and the population size. We conclude that equipping agents with inertia and trend-seeking is key to simultaneously produce (i) the macroscopic features of delay and explosiveness regularly observed in real-world social diffusion[1,3,5,7,13,14,17], and (ii) generate individual-level responses consistent with our experimental data and the psychology literature. Further investigation reveals that the length of delay, which determines whether diffusion occurs or not, is critically shaped by the composition of the population, viz. the fraction of committed minority and/or of explorers in the population. While diffusion is guaranteed above a critical threshold of 25% committed minority, in agreement with Centola et al.[10], we expand on this by showing that when the committed minority is below the 25% threshold, social change can still occur if the rest of the population is sufficiently sensitive to trends.

## Results

**Experimental evidence**. The social psychology literature discussed in the Introduction establishes that inertia and trend-seeking have a key role in individual decision-making, across a range of scenarios. We conducted a set of online multi-round

game experiments to study social diffusion leading to change in conventions. Importantly, the experiment provided individual-level data which was used (i) as additional motivation, besides support from the social psychology literature, to incorporate inertia and trend-seeking mechanisms within our ABM, (ii) for parametrisation of our model, and (iii) to illustrate how existing models lacking inertia and trend-seeking produce unrealistic individual-level decision-making patterns.

In our experiment, 180 recruits were enrolled and divided into 20 small groups (each group with 8–10 recruits) and participated in a multi-round game. Of the 180 recruits, 32 dropped out after enrolling but before participating in the game, while the other 148 participated in the game (full details on the experimental setup can be found in the 'Methods'). In each round, participants were asked to choose between two strategies, and were able to see the proportion of the rest of the group that chose each of the two strategies in the previous round, but no information was provided as to who selected which strategy. The game ended when all participants in the group selected the same strategy in the same round, reaching thus a full consensus, or after 24 rounds if no consensus was reached. If a full consensus was reached, we called that strategy the winning strategy. Besides a base reward for participation, a group monetary reward was available to be split among the participants if and only if a full consensus was reached, promoting coordination and consensus-seeking. The group reward decayed over time, while a participant's share of the group reward was proportional to how often he or she chose the winning strategy.

The monetary reward was designed to capture several aspects of real-world social conventions; the central requirement is to reach consensus, with merits to doing so quickly, but also benefits for converting others to your choice. To replicate a real-world diffusion process leading to a change in the social convention, we included into each group 2 (17%), 3 (25%) or 4 (33%) computer bot players with pre-programmed strategies, called committed minority bots, so that each group has 12 players in total (humans and bots). In Stage I of the game, the bots helped to establish a status quo: the strategy that all regular participants and all but one bot adopt in the same round. Stage II began in the following round; the committed minority bots changed role to stimulate a diffusion process by stubbornly choosing the alternative (non-status quo) strategy until the game ended.

The results of our experimental study are summarised in Fig. 1. Figure 1a shows the proportion of participants (i.e. excluding computer bots) that adopt the alternative strategy across the rounds in the 20 experimental trials; three of them are highlighted as representative of fast diffusion (green), delayed diffusion (blue), and no diffusion (red). The whole experimental data are available in an online repository[46], and reported in Supplementary Fig. 2 and Supplementary Table 1 with additional details. See also the 'Data availability' statement. In 18 of the 20 trials, the status-quo strategy is established within 1–3 rounds (which marked the end of Stage I). In the majority of the trials (16 of the 20 trials), full diffusion to the alternative strategy occurs at some point in the experiment. In 15 of the trials which saw diffusion occurring, the diffusion was explosive, irrespective of the number of committed minority bots in the group (as in the green and blue trials in Fig. 1a). This explosiveness was present even in the three groups in which the take-off time was sensibly delayed (see Fig. 1a, blue curve); in these groups, the status quo strategy remained adopted by the large majority of the group for several rounds, before a rapid diffusion of the alternative occurred.

The switching rates $y_v$ for the 148 individuals who participated in the games are reported in Fig. 1b. We will formally define $y_v$ in the sequel after introducing the ABM, but roughly speaking, $y_v$ records the fraction of rounds in which player $v$ changed strategy over the duration of the game. Figure 1b suggests that the switching activity is in general moderate (on average, one switch every 14 rounds) and highly heterogeneous. There is a large peak in the distribution that comprises 99 players (67% of the players) who have $y_v \leq 0.046$. In the context of the game, this typically meant (i) the player switched only in the very last round that resulted in a consensus being reached on the alternative strategy, or (ii) no diffusion occurred, and the player switched at most once over the 24 rounds of the game, or (iii) the player switched to the alternative strategy while it was selected by a minority of the group, and did not change strategy after. The remaining 49 players have a wide distribution of switching rates, typically switching several times before the game ended. These remaining players showed a greater propensity to switch away from the status quo towards the alternative strategy, even when the status quo was the overwhelming majority strategy. The nature of the aforementioned heterogeneity suggests that players (i.e. not committed minority bots) can be classed either as explorers who are willing to try the alternative despite the lack of majority support, or non-explorers who tend to adopt the alternative once it is clear it will be the final winning strategy. This heterogeneity is also consistent with existing diffusion literature, with individuals

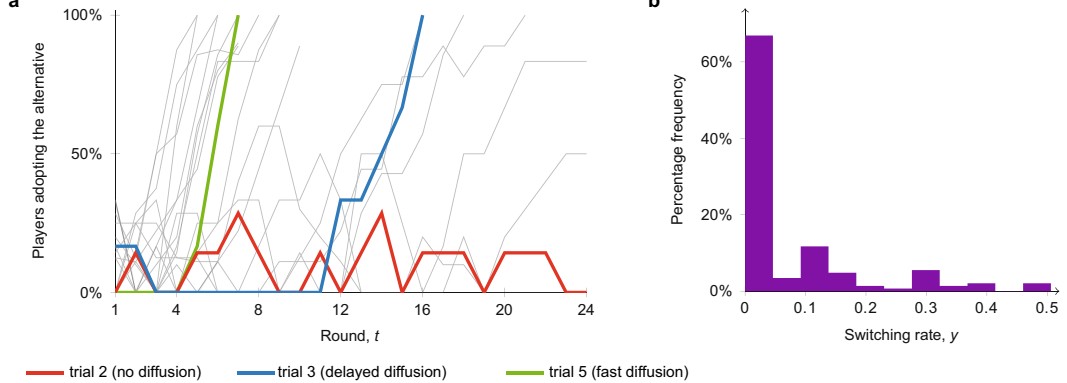

**Fig. 1 Population-level and individual-level experimental results. a** Temporal evolution of the fraction of participants adopting the alternative in all the 20 trials starting from Stage I. Three trials are highlighted; with committed minority bots making up 25% of the group, the trials illustrate fast diffusion (green), delayed diffusion (blue), and no diffusion (red), respectively. **b** Distribution of the empirical switching rates of the participants (the fraction of rounds a participant switched strategy during the trial). Notice the highly heterogeneous distribution, with a large peak close to 0 (containing around 67% of all players), and the remainder widely distributed. In the sequel, we show that this type of distribution arises in ABMs when agents are impacted by both inertia and trend-seeking.

**Table 1 Regression table displaying the effects of the three predictors—(fraction of) adopters in previous round, choice in previous round, and group trend—on the dependent variable choice in current round.**

|  | Estimate | t-statistic | p | 95% conf. interval |
|---|---|---|---|---|
| Intercept | 0.073 | 6.21 | <0.001 | [0.050, 0.096] |
| Adopters in previous round | 0.53 | 15.83 | <0.001 | [0.46, 0.59] |
| Choice in previous round | 0.36 | 15.95 | <0.001 | [0.31, 0.40] |
| Group trend | 0.31 | 6.10 | <0.001 | [0.21, 0.41] |
| $R^2$ | 0.46 |  |  |  |
| F-score | 494.66 |  | <0.001 |  |

typically classified respectively as either early adopters/early majority or laggards/late majority[21]. A systematic method for classifying players as explorers or non-explorers was finally developed, leading to the identification of 85 explorers and 53 non-explorers (see 'Methods').

We conclude by analysing the individual-level data to strengthen the motivation for inclusion of inertia and trend-seeking mechanisms in our ABM, which is proposed in the next section. First, we performed a regression analysis with fixed effects estimator[47] (see Supplementary Note 2 for details and full results). As reported in Table 1, the analysis supports the presence of coordination, inertia and trend-seeking in the individual-level data (F-score = 494.66, $p < 0.001$). Specifically, the fraction of adopters in the previous round, the individual's choice in previous round, and the group trend explains 46% of the over-time variance in the individual's choice in current round ($R^2 = 0.46$), and all three predictors are statistically significant ($p < 0.001$). Interestingly, besides confirming the key role of coordination, these findings highlight the crucial role of inertia and trend-seeking. In fact, we observed that if an individual has chosen a given strategy in the round immediately prior, the probability that he or she will choose it again in the current round increases by 36%. If the whole group except the given individual switched strategy in the previous round (thus generating a trend), then the probability that the considered individual will also choose the trending strategy in the current round increases by 31%.

We also conducted two additional tests on the individual-level data, focusing separately on inertia (Wald–Wolfowitz runs test, $p < 0.0001$) and trend-seeking (binomial test, $p \approx 0.005$)[48]. Full details are reported in Supplementary Note 2. In summary, the regression analysis and additional tests examining the experimental data at the individual level provided support for the presence and the significance of inertia and trend-seeking in individual-level decisions that participants made during our experimental game and, consequently, in creating the shape of the diffusion patterns observed in Fig. 1. Building on the existing literature[40–42,44,45], the reported findings further motivate the inclusion inertia and trend-seeking—together with social coordination—as the key mechanisms underlying each agent's decisions in the ABM.

**Agent-based model**. The existing literature provided motivation to include inertia and trend-seeking in models of social diffusion, and this was further strengthened by the analyses performed on our experimental data. We now incorporate these two behaviours into an ABM so we can examine the societal-level consequences of these individual-level mechanisms. The proposed mathematical model falls within the framework of game theory[49], which has become a widely adopted paradigm for representing complex

decision-making processes during social diffusion[31,32,36,38]. A key advantage of the framework lies in the possibility of encapsulating specific mechanisms of the studied decision-making process by adjusting the payoff function of the game. In this paper, we modify a standard payoff function for coordination games, used in the literature to model convention change and social diffusion[32,36,38], to incorporate inertia and trend-seeking while retaining unchanged the fundamental game-theoretic decision-making framework.

The game is played by a set $\mathcal{V} = \{1, \ldots, n\}$ of $n \geq 2$ players. Rounds are denoted by discrete time-steps $t = 1, 2, \ldots$. Each player $v \in \mathcal{V}$ can choose between two strategies from the set $\mathcal{S} = \{0, 1\}$, where 0 represents the status quo and 1 represents the alternative. The strategy played by player $v$ at time $t$ is denoted by $x_v(t) \in \mathcal{S}$. At each time-step, players are allowed to revise their strategy, using a revision process following a noisy best-response rule[50]. Specifically, the probability that player $v$ adopts strategy $x \in \mathcal{S}$ at time $t + 1$ is computed through the log-linear (logit) learning formula as

$$\mathbb{P}[x_v(t+1) = x] = \frac{\exp\{\beta_v \pi_v(x)\}}{\exp\{\beta_v \pi_v(0)\} + \exp\{\beta_v \pi_v(1)\}}, \quad (1)$$

where $\beta_v \geq 0$ is a measure of the rationality of individual $v$ in the decision-making process: for $\beta_v = 0$, strategies are revised fully at random, and for $\beta_v = \infty$, Equation (1) reduces to a deterministic best-response rule. The function $\pi_v(x)$ is the payoff of player $v$ for adopting strategy $x \in \mathcal{S}$, which is given by

$$\pi_v(1) = \frac{b_v}{n-1} \sum_{w \in \mathcal{V} \setminus \{v\}} x_w(t) + k_v x_v(t) + r_v \hat{x}_v(t), \quad (2a)$$

$$\pi_v(0) = \frac{b_v}{n-1} \sum_{w \in \mathcal{V} \setminus \{v\}} \left(1 - x_w(t)\right) + k_v\left(1 - x_v(t)\right) + r_v\left(1 - \hat{x}_v(t)\right), \quad (2b)$$

where $b_v, k_v, r_v$ are non-negative scalar constants and

$$\hat{x}_v(t) = \frac{1}{2}\left[1 + \frac{1}{n-1} \sum_{w \in \mathcal{V} \setminus \{v\}} \left(x_w(t) - x_w(t-1)\right)\right]. \quad (3)$$

We impose $b_v + k_v + r_v = 1$ for all $v \in \mathcal{V}$, so that the payoff defining the decision-making process is a convex combination of three separate summands on the right-hand-sides of the payoff functions in Equation (2a) and (2b). The first summand is a standard coordination game mechanism, which captures social coordination among agents[32,36]: the more other players are playing strategy $x \in \mathcal{S}$ at time $t$, the higher the payoff for choosing that strategy. Importantly, our proposed extension of this standard coordination model occurs through the inclusion of two further summands. The second summand captures inertia: player $v$ increases their payoff by $k_v$ for sticking with their current strategy. The third summand encapsulates the trend-seeking process. In fact, the quantity $\hat{x}_v(t) > 1 - \hat{x}_v(t)$ if and only if the fraction of adopters of $x = 1$ has increased in the previous time-step. Hence, the third term provides an increased payoff for playing strategy $x \in \mathcal{S}$ whenever the fraction of adopters of $x$ has increased in the previous time-step.

It is worth remarking that further features and mechanisms may be included in the model. For instance, the exchange of information between players can be driven and restricted by a network of interactions[32,36]. More sophisticated terms to model inertia and trend-seeking mechanisms can be designed to capture, for example, long term memory and a decreased impact of trends once a strategy is in the majority, respectively. However, for the purposes of this paper, we consider a minimalistic implementation of the model with all-to-all communication between players, synchronous updates, and a simple formulation for inertia and

trend-seeking. As we will see in the sequel, this implementation enables us to isolate and examine how inertia and trend-seeking shapes diffusion, and still allows for capturing fundamental characteristics of real-world diffusion processes, without the confounding effects that may be caused by the introduction of further complex features.

**Diffusion driven by a committed minority**. To stimulate social diffusion, we introduced a small set of committed minority $\mathcal{C} \subset \mathcal{V}$, who stubbornly play the alternative strategy throughout the game; this is achieved by setting $k_c = 1$, $\beta_c = \infty$, and $x_c(0) = 1$, for all $c \in \mathcal{C}$. All the other agents, termed regular, start by playing the other (status quo) strategy, i.e. $x_v(0) = 0$, for all $v \in \mathcal{V} \setminus \mathcal{C}$. This setup thus replicates our empirical study beginning at Stage II—i.e. the moment when all the committed minority bots start choosing the alternative.

The stochastic process induced by Equation (1) ensures that every strategy configuration of the population occurs with non-zero probability, which implies that the configuration where all agents play the alternative strategy, representing full diffusion, will be reached with probability 1. Thus, the key question is not whether diffusion will take place, but how quickly does full diffusion occur? Toward that end, we define the following measures to study diffusion processes and evaluate its characteristics. The diffusion time,

$$T^* := \inf\left\{ t \geq 0 : \frac{1}{n}\sum_{v \in \mathcal{V}} x_v(t) \geq 0.99 \right\}, \tag{4a}$$

quantifies the time needed for the alternative strategy to spread across the entire social group. The take-off time,

$$\bar{T} := \sup\left\{ t \leq T^* : \frac{1}{n}\sum_{v \in \mathcal{V}} x_v(t) \leq 0.4 \right\}, \tag{4b}$$

is the time required for the diffusion process to reach a critical threshold and take off, ensuring a transition toward the alternative. Note that at $\bar{T}$, the alternative is still in the minority, even accounting for the presence of committed minority, and a large value of $\bar{T}$ is evidence of delayed diffusion. Supporting motivations for setting the threshold value at 0.4 in Equation (4b) and robustness checks for our findings under different settings can be found in the Supplementary Note 4 and Supplementary Figs. 8–11. The transition time,

$$\Delta T := T^* - \bar{T}, \tag{4c}$$

measures the explosiveness of the diffusion process: the smaller the $\Delta T$, the sharper the adoption of the alternative. Together, the pair $(\Delta T, \bar{T})$ characterises the diffusion process, as described in the Methods. Finally, for any $v \in \mathcal{V} \setminus \mathcal{C}$, the switching rate

$$y_v = \frac{1}{T^*}\left( \sum_{t=1}^{T^*} \left| x_v(t) - x_v(t-1) \right| - 1 \right), \tag{4d}$$

as anticipated in the prequel, counts the normalised number of times agent $v$ has revised its strategy, up to the diffusion time $T^*$, and excluding the final revision.

By employing Monte Carlo numerical simulations of the proposed model, we are able to examine factors that cannot be easily studied in an experimental set up due to practical limitations. We first focus on unveiling the role of inertia and trend-seeking in shaping collective patterns of diffusion in large-scale populations. Then, we examine how the composition of the population in terms of committed minority and explorers affects social diffusion.

To begin, we used the experimental data to parametrise the values of $b_v$, $k_v$, and $r_v$, which are the weights in Equation (2a) and

(2b) associated with the social coordination, inertia, and trend-seeking mechanisms, respectively. Specifically, we identified two classes of regular players: explorers and non-explorers, with parameters equal to $b_e = 0.48$, $k_e = 0.10$, and $r_e = 0.42$; and $b_f = 0.42$, $k_f = 0.42$, and $r_f = 0.16$, respectively. Hence, all three mechanisms play a role in the decision-making of agents from both classes. Coordination is equally important for both classes, and while explorers are more affected by trend-seeking than inertia, it is the opposite for non-explorers. Assuming that rationality is the same for both explorers and non-explorers, parametrisation yielded $\beta_v = 7.8$ for all participants. We define the fraction of committed minority in the population as $|\mathcal{C}|/n$, and we denote the fraction of explorers among the remaining, regular agents as $\rho_e$. More details on the model parametrisation and the simulation setup can be found in the Methods.

**Inertia and trend-seeking yield realistic diffusion patterns**. Our analysis starts by observing in Fig. 2 the time evolution of the model for two different fractions of explorers, $\rho_e$, with 50 simulation runs each. We draw attention to three features of the diffusion process. At the macroscopic level, we observe the two salient features of delayed take-off and explosive diffusion, which are consistent with the experimental results and real-world empirical observations of social diffusion over time. The stochastic nature of the model implies the take-off time $\bar{T}$ differs between the simulation runs, but it is evident that $\bar{T}$ increases as the fraction of explorers, $\rho_e$, decreases. This is because non-explorers have a greater inertia and are less susceptible to trends. However, the diffusion is always explosive once the process takes off, with $\Delta T$ being small even if there is a significant delay before take-off ($\bar{T}$ is large). The third salient feature is observed at the microscopic level, concerning the switching activity of the agents. We observe that the distribution of $y_v$ is strongly heterogeneous, and the switching rate is moderate, with on average one switch every 17 rounds for $\rho_e = 0.2$ and every 11 rounds for $\rho_e = 0.6$. Indeed, the shape of the switching rate distributions resemble the empirical distribution in Fig. 1b, whereby there is a large peak around 0 and the remaining rates are broadly distributed between 0 and 0.3. The individual-level decision-making patterns generated by means of the proposed model are thus qualitatively consistent with the experimental data, and capture the different behaviours of explorers and non-explorers.

To illustrate the importance of these findings, we next show that existing ABMs based only on the social coordination mechanism[32,36,38] are not able to capture the three aforementioned microscopic and macroscopic features simultaneously. To begin, we use the same parametrisation process (as detailed in 'Methods') to calibrate a model with only social coordination, that is, by enforcing $b_v = 1$ and $k_v = r_v = 0$. By allowing for different levels of rationality for explorers and non-explorers, we obtain $\beta_e = 4.8$ for explorers and $\beta_f = 19.7$ for non-explorers. The simulations of the model obtained are depicted in Fig. 3. At the macroscopic level, no diffusion is observed due to the high rationality of the non-explorers preventing switching of strategy at the microscopic level (Fig. 3c and d). Realistic macroscopic diffusion patterns can be observed if the ratio of explorers is increased to $\rho_e = 0.7$ (Fig. 3e) or the rationality of non-explorers is reduced to $\beta_f = 8.5$ (Fig. 3g). However, the microscopic features in these scenarios (Fig. 3f and h) differ significantly from the empirical data (Fig. 1b), with a higher and less heterogeneous switching rate. In the Supplementary Note 6 and Supplementary Figs. 14–15, we show that including just inertia or just trend-seeking is not sufficient to produce the desired diffusion patterns. When both inertia and trend-seeking are included however, the macroscopic and microscopic features of social diffusion are

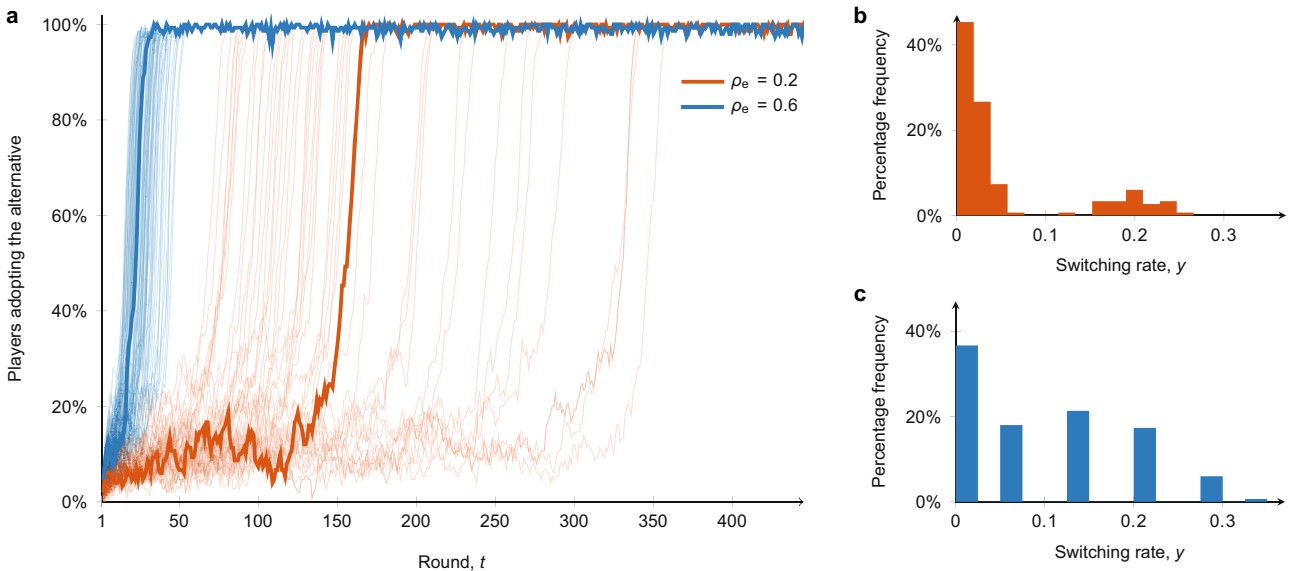

**Fig. 2 Sample simulations of the ABM with two different fractions ($\rho_e$) of explorers relative to non-explorers.** Simulations are performed with $n = 200$ agents and 25% of committed minority. **a** Temporal evolution of the fraction of regular agents adopting the alternative with $\rho_e = 0.2$ (orange) and $\rho_e = 0.6$ (blue) in 50 simulation runs for each one of the two values of $\rho_e$; two representative simulations are highlighted. **b** Switching rates of the regular agents for $\rho_e = 0.2$ in a representative simulation. **c** Switching rates of the regular agents for $\rho_e = 0.6$ in a representative simulation.

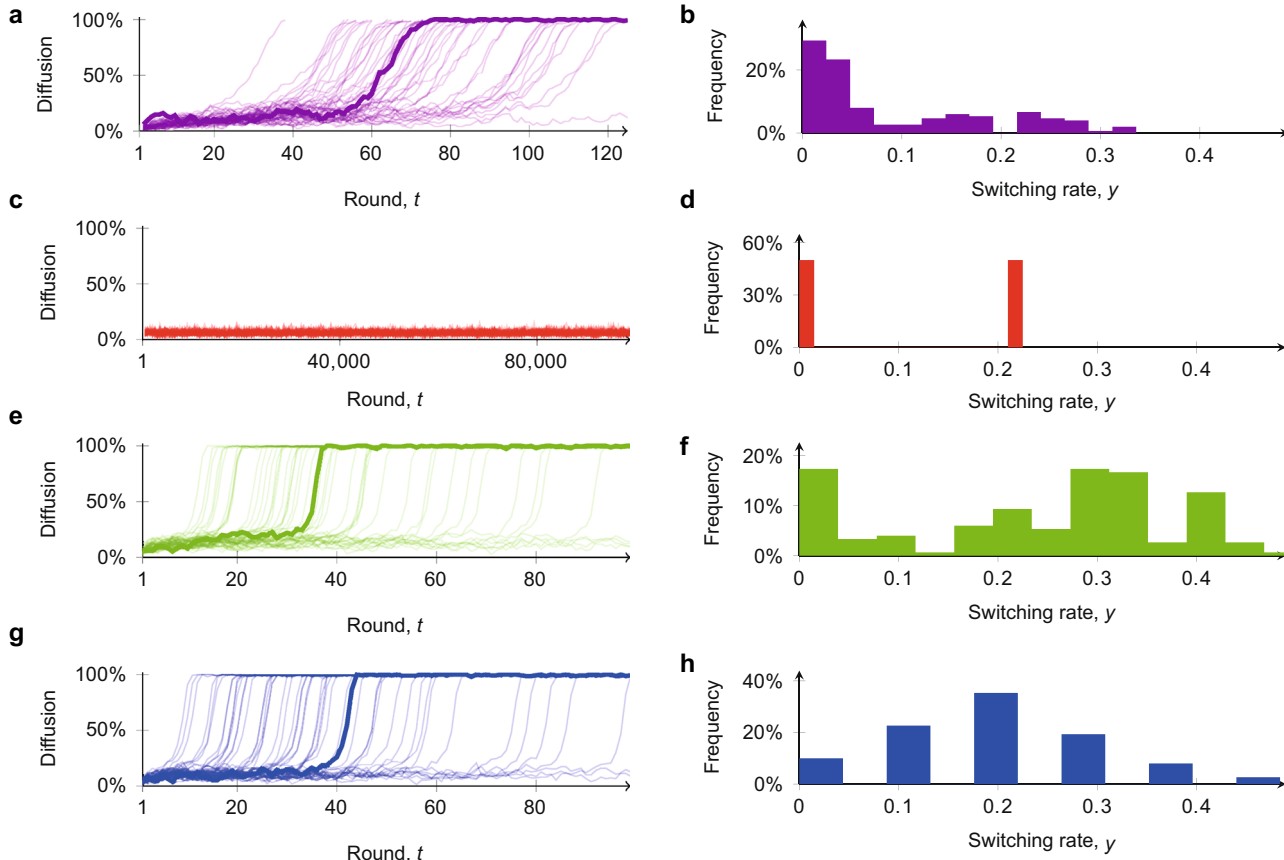

**Fig. 3 Comparison of the proposed agent-based model with a standard coordination game model.** In all figures, we set $n = 200$, 25% of committed minority, and we generate 50 independent simulation runs. One representative simulation is highlighted in each plot, with the corresponding switching rate in the right panel. **a, b** Simulations of our model with $\rho_e = 0.5$. **c, d** Simulations of a pure coordination model calibrated from our empirical data ($\beta_v = 4.8$ for explorers and $\beta_v = 19.7$ for non-explorers), with $\rho_e = 0.5$. **e, f** Explosive delayed diffusion is obtained with a pure coordination game by increasing the fraction of explorers to $\rho_e = 0.7$. **g, h** Explosive delayed diffusion is obtained with a pure coordination game by decreasing the rationality of non-explorers to $\beta_v = 8.5$.

simultaneously captured (Fig. 3a and b), highlighting the crucial and nontrivial interplay between the two mechanisms and the presence of both as a necessity. Further comparisons with other ABMs, such as epidemic models and linear threshold models, are reported in the Supplementary Note 5 and Supplementary Fig. 13.

Explosiveness of the diffusion process, possible delays before diffusion takes off, and a moderate and heterogeneous individual-level switching activity are features that characterise the emergent behaviour of our model. These same features are common to many real-world diffusion processes[3,13,14] (see Supplementary Note 7 and Supplementary Fig. 16), and are consistent with our experimental observations (Fig. 1) and with the social psychology literature on inertia[40,43]. A central conclusion from the above analysis of our model is thus to demonstrate that including inertia and trend-seeking into the decision-making process of each agent in the ABM is critical to ensure the simultaneous capturing of both individual and population-level features of social diffusion. Without this explicit inclusion, existing models, besides being inconsistent with the social psychology literature, cannot reconcile the individual and population-level outcomes.

**Emergent behaviour in large populations**. To better elucidate on the three key features of diffusion processes of (i) delayed take-off time, (ii) explosive diffusion and iii) moderate and heterogeneous switching activity, we put forward a campaign of Monte Carlo simulations, varying the population size $n$ and fraction of explorers $\rho_e$. In these simulations, we set 25% of committed minority (corresponding to the experimental trials with 9 participants and 3 committed minority bots), with the results summarised in Fig. 4. Supplementary Note 5 and Supplementary Figs. 3–7 record additional results examining the robustness of our findings for different sets of parameters and for different fractions of committed minority.

At the macroscopic level, numerical simulations in Fig. 4a yield several striking observations. The first elucidates the role of inertia and trend-seeking in determining the diffusion pattern: there is always a delay before take-off occurs, and the delay $\bar{T}$ increases as the fraction of agents highly susceptible to inertia (non-explorers) increases. Second, the importance of explorers in unlocking social diffusion is crystallised. In the absence of explorers, i.e. $\rho_e = 0$, the take-off time $\bar{T}$ grows greater than linearly with respect to the population size $n$. In contrast, the presence of a sufficiently large fraction of explorers, with $\rho_e > 0.05$, leads to take-off time $\bar{T}$ that is moderate and independent of population size $n$. In other words, the presence of explorers (who have low inertia but are highly affected by trends) may be necessary for diffusion to occur because the resulting delay in their absence in large communities effectively prevents the committed minority from kick-starting social diffusion. This independence is similar to that reported in the literature for a different model of diffusion[36]. Further in the next section, we will deepen this analysis showing that, in fact, such independence is determined by a combination of the fraction of committed minority and explorers.

The results summarised in Fig. 4b show that the transition time $\Delta T$ is always small, independently of the population size. This property is verified even for $\rho_e = 0$, where the take-off time $\bar{T}$ may grow greater than linearly as population size increases, but the explosiveness of the diffusion phenomena is retained. This suggests that explosiveness is an inherent feature of the proposed model, attributable to the trend-seeking mechanism, which is also present but with reduced intensity in the decision-making processes of non-explorers.

Finally, by observing the outcome of the simulations at a microscopic level in Fig. 4c, we register a moderate switching activity with a heterogeneous distribution. Both the heterogeneity,

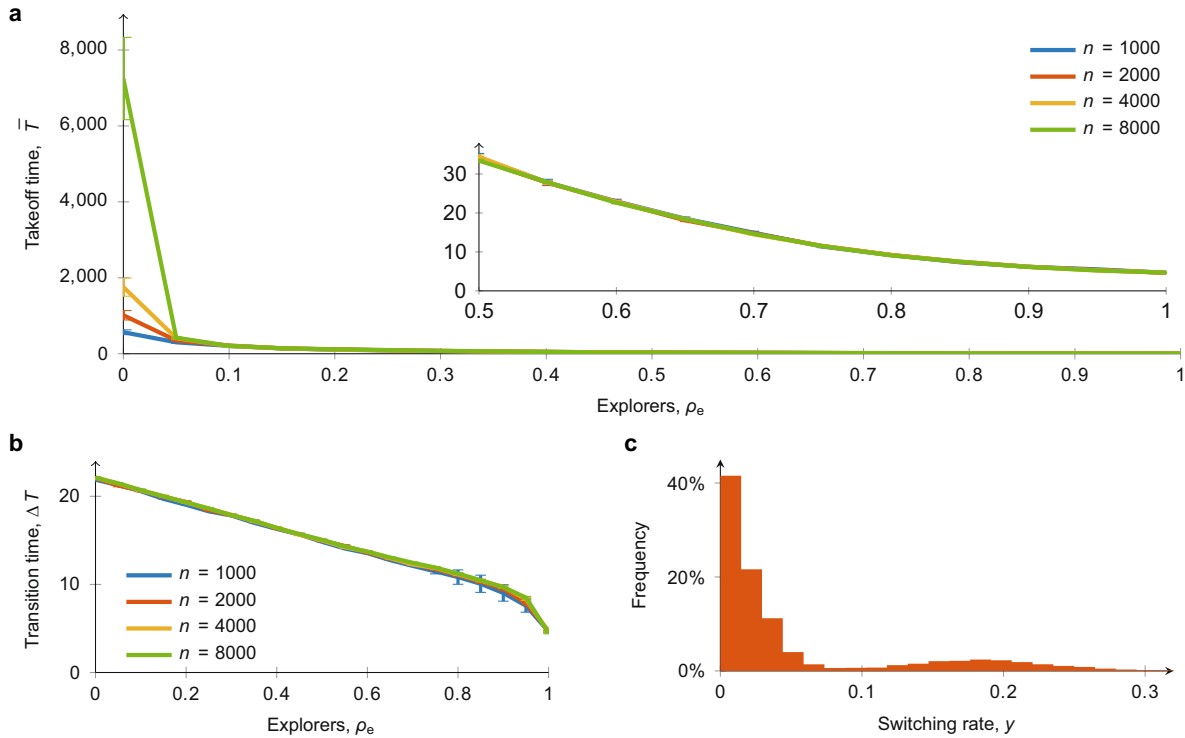

**Fig. 4 Monte Carlo estimation of the take-off time, the transition time, and the switching rate of regular agents.** Estimations are performed over 200 independent simulations. **a** Take-off times ($\bar{T}$) estimated for different fractions of explorers and increasing population sizes, with a fixed 25% committed minority. Vertical bars are 95% confidence intervals. **b** Transition times ($\Delta T$) estimated for different fractions of explorers and increasing population sizes, with a fixed 25% committed minority. Vertical bars are 95% confidence intervals. **c** Switching rate ($y$) estimated for $n = 1000$ and $\rho_e = 0.2$.

which captures the difference in decision-making patterns between explorers and non-explorers, and the moderate switching levels, which is due to the presence of inertia, closely match our experimental data in Fig. 1b. Outcomes for 17% and 33% of committed minority are reported in Supplementary Note 5.

**Key role of committed minority and explorers.** Our final line of analysis examines the role of the committed minority and explorers in determining the delay in the take-off and, ultimately, whether or not diffusion can practically occur in the real world. In our simulations for this subsection, we considered a fixed population of $n = 1000$, and recorded the delay $\bar{T}$ while varying the explorer to non-explorer ratio $\rho_e$ and the fraction of committed minority in the population, $|\mathcal{C}|/n$. The simulation stopping time was set at 50,000 time steps, meaning that $\bar{T} > 50,000$ if no diffusion was observed in the simulation window.

As Fig. 5 shows, when the population has more than 25% of committed minority, the delay $\bar{T}$ increases as the fraction of explorers $\rho_e$ decreases, but it is always small ($\bar{T} < 150$) and independent of the population size (consistent with Fig. 4a). When the committed minority comprise less than 19% of the population, no diffusion is observed within the simulation window of 50,000 time steps, irrespective of the fraction of explorers. Interestingly, there exists a critical regime between 19–25% of committed minority, in which the fraction of explorers appears to play a critical role in determining the size of the delay. Thus, for any given fraction of explorers $\rho_e$, there exists a committed minority fraction $|\mathcal{C}|/n$ that guarantees diffusion will occur. The converse is not true however; when $|\mathcal{C}|/n$ is below 0.19, there are no values of $\rho_e$ for which diffusion is observed in the simulation window.

Our findings corroborate with Centola et al.[10], who identified that a committed minority comprising 25% of the population is a sufficient condition to trigger social change, but also provide several important and additional conclusions. First, the model predicts that diffusion will never be observed in real-world scenarios if there is less than ≈19% committed minority, because irrespective of the fraction of explorers, the take-off time $\bar{T}$ is so large that the committed minority will die out, a newer alternative appears, or other exogenous changes occur in the population before the diffusion process takes off. Second, we find that in an intermediate critical regime below the 25% threshold and above 19% committed minority, the presence of enough explorers may unlock diffusion by significantly reducing the delay. Interestingly,

there appears to be a sharp phase transition in unlocking diffusion in the critical regime: when the fraction of explorers decreases below a threshold value dependent on the fraction of committed minority, the delay sharply blows up from in the order of 10–100 rounds to greater than 50,000 rounds. Our findings thus highlight the importance of explorers in unlocking social change if the committed minority do not reach the 25% critical mass (as they may be marginalised[51]).

**Discussion**

Motivated by the existing literature showing that inertia and trend-seeking are two mechanisms that can significantly influence individual decision-making, we have proposed a mathematical model that generalises the coordination game, which is a popular diffusion and decision-making framework for ABMs. Through the analysis of the experimental data and the conclusions drawn from extensive simulations of the proposed model, we have highlighted how inertia and trend-seeking play key roles in shaping patterns of social diffusion, determining macroscopic features such as the delay before diffusion take-off and the explosiveness of the transition process. At the individual-level, our experiment and simulations dovetail with existing literature on the importance of a committed minority in overturning social conventions[10], and went further to illustrate the crucial role played by explorers in unlocking social diffusion. Future research may explore the relation between the take off of a diffusion process and its explosiveness with the well-known concept of a tipping point which has been explored in the diffusion literature[10,15,20,52].

Population models, such as the Bass model, under certain parametrisations are able to describe macroscopic-level features of diffusion detailed in this work, such as explosiveness and delay before diffusion take-off[16,19,20]. However, the inherent population-level description of the phenomenon that is adopted in these models prevented researchers from directly incorporating and studying the impact of the individual-level mechanisms of inertia and trend-seeking, as well as to capture the high heterogeneity across the individuals (which is observed in the switching rates recorded in our experimental data). In taking an agent-based approach driven by game theory, we hope to have underlined the importance of incorporating the behavioural mechanisms of inertia and trend-seeking into existing models to capture realistic features of social diffusion phenomena at both the microscopic (individual) and macroscopic (societal) levels simultaneously. Inertia and trend-seeking are likely to be pervasive in many collective decision-making scenarios, such as joining social justice movements[53] or adoption of sustainable practices[54], and generalisations of our model to other problem settings, beyond the setting of convention change, may be of interest. In the following, we discuss a few possibilities.

We considered the mechanisms of inertia and trend-seeking in a social diffusion model framed in terms of social conventions where the status quo and alternative have equal benefits, but more general collective decision-making scenarios can be investigated. For instance, in the diffusion of innovation literature, the novel alternative often has a clear benefit over the status quo (such as hybrid corn seed offering better crop yield[55,56]). This can be incorporated into our ABM by adding a payoff advantage to one of the two strategies, which also ties our framework with the social learning literature[38,57].

It has been suggested that cultural differences may impact the likelihood or ease of securing changes in social conventions[58]. Parametrisation of the model involved data from recruited individuals who were native English speakers, and it would be of interest to conduct our study with individuals recruited from

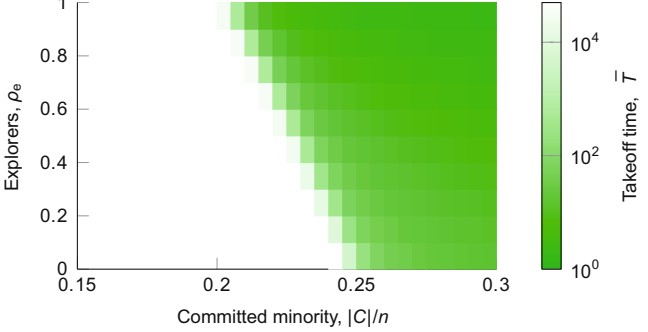

**Fig. 5 Monte Carlo estimation of the take-off time for different committed minorities ($|\mathcal{C}|/n$) and fractions of explorers ($\rho_e$).** The heat map identifies the delay in the take-off time ($\bar{T}$) averaged over 200 Monte Carlo simulations for a given fraction of committed minority ($|\mathcal{C}|/n$), and of explorers ($\rho_e$), for a population of size $n = 1000$. Each simulation stops at $t = 50,000$ steps, so that white indicates that the delay in the take off $\bar{T} > 50,000$, on average. Note the logarithmic scale of the heat map.

different cultural backgrounds. Different individual-level characteristics such as a higher propensity to conform may alter key model parameters, and generate different macroscopic-level diffusion predictions. Our work deliberately assumed a fully connected (all-to-all) population interaction structure, in order to isolate and highlight the role of inertia and trend-seeking in the diffusion process. In the real world, diffusion often occurs over social networks with non-trivial structures, which can either favour or hamper diffusion[32]. Additional investigations on the role of network structures for the proposed model should be conducted. Moreover, an extension of the model to incorporate different interaction mechanisms such as asynchronous, pairwise, or time-varying interactions should be investigated, toward utilising the proposed ABM in different experimental and real-world settings, such as the one considered by Centola et al.[10]. By expanding our model to encompass three or more strategies[59] and considering committed minority supporting several alternatives to a status quo, a richer study of how inertia and trend-seeking may shape the evolution of societal conventions can be pursued.

In summary, we have considered a simplified model whereas real-world social diffusion clearly has many complicating factors that should be investigated in future studies. Nonetheless, we believe our model is general enough, and amenable to extensions, to be of interest to a broad range of researchers from different scientific communities working on social diffusion. Through theoretical and empirical approaches, a growing body of literature on complex contagion[8,27–30] has explored the complexities of the individual-level dynamics that govern social diffusion. We hope our work, centring on theoretical modelling supported by empirical evidence, has provided a strong argument for further close study of behavioural mechanisms in diffusion models. These mechanisms are characteristic features of human decision making[40–45], and result in social diffusion with complex contagion characteristics that distinguish it from other spreading and contagion processes like epidemics, biological evolutionary dynamics, and computer viruses.

## Methods

**Experimental setup**. We enrolled 180 recruits through the Prolific Academic platform (https://www.prolific.co) for our experimental study. There were 71 female, 75 male, and 34 recruits did not provide their gender. Their average age was 31 years, ranging from 18 to 76. A majority of our sample (113 recruits) completed some form on college or held a university degree, while 35 completed high school or lower equivalent and 32 did not provide their education level. The Prolific Academic platform selected participants who were all native English speakers.

The game was programmed using the oTree platform and played online[60]. Before entering the study, recruits were informed that their participation in the study is voluntary, that they can discontinue it at any point without repercussions, and were asked to consent to these terms in order to begin their participation. Upon entering the study, recruits first received a set of instructions and questions checking their understanding of the game; 32 recruits failed to complete the instructions procedure and did not participate in the actual game. These dropouts were replaced with a dropout computer bot (see below for details), in order to maintain the group size. Then, they played the experimental game in groups of 8–10 human participants; special committed minority computer bots were introduced in a quantity that ensured the game always had 12 players total (the role of the committed minority bots, which differ from the dropout bots, will be explained below). The participants in our experiment are those 148 individuals who completed the instructions procedure. At the end, participants provided some demographic information and were debriefed about the full contents of the study. All of the pages in the experiment were timed, so that the game continued even if certain participants dropped out for one, or multiple rounds. Each experimental session lasted typically 20–30 min.

The participants were informed that they were members of a company board, who were voting on which of two products, Eta or Tao, the company should invest in. It was explained to the participants that in order for an investment to be made on a product, it was required to be supported by all members of the board, i.e. a unanimous consensus vote. In the game, this translated to the following mechanism. In each round of the game, participants were first asked to choose between two strategies, corresponding to investing in one of two products, Eta or Tao. After making the choice, each participant could see the proportion of others in the group, including all the bots, that chose each of the two strategies (products) in the given round. However, the identity of which participants (including bots) selecting which strategy were not revealed. Each participant could then revise his or her strategy in the next round. The game ended when all participants selected the same strategy, i.e. a consensus, or after 24 rounds if no consensus was reached (which represented the opportunity to invest in either product being lost in the game, and relates to the cost of failing to adopt the same option in social conventions). Note that our experimental setup is inspired by the one used by Centola et al.[10], and coordination is central to the payoff system in both setups. However, the interaction mechanisms differ: our participants engage in all-to-all interactions and observe information about the strategy choices of the entire group at each round, while Centola et al. considered pairwise interactions that were randomised in each round, through which individuals at each round can gather information on only a single peer. Both mechanisms can occur in real-world social conventions: pairwise interactions may occur for greetings (handshakes vs bowing) while group interactions may occur when people decide to walk on the left or right side of the footpath.

In each group, we included committed minority computer bots, whose decisions were pre-programmed in two stages. In Stage I, at the beginning of the game, all but one committed minority bot chose the majority-supported product to help drive towards an initial consensus among the human players, to stimulate a natural forming status quo. Stage I ended when all human players and all but one of the committed minority bots have selected the same strategy in the same round, which we term the status quo strategy; this corresponds to strategy $x = 0$ in the ABM. Because a single committed minority bot is selecting the non-status quo strategy at this point, there cannot be a consensus during Stage I. Stage II began in the round immediately after Stage I ended, that is, in the round where all (human) participants selected the status quo for the first time. In Stage II, all committed minority bots then chose the alternative strategy, different from the status quo strategy and stubbornly continued to do so until the game ended with full consensus on the alternative strategy or after 24 rounds and no consensus; the alternative strategy corresponds to strategy $x = 1$ in the ABM. We included 4, 3, and 2 committed minority bots into groups with 8, 9, 10 human participants, respectively, to ensure that each group had 12 players. The decisions of the committed minority bots were included in the strategy distribution observed by the participants at the end of each round. If a human participant did not choose a strategy in any given round, i.e. he or she dropped out before the game began or was too slow in choosing the strategy, he or she was replaced by a computer bot in that given round, who selected the strategy chosen by a majority of other players.

A participant's monetary reward was the sum of a base reward of £3.50 and a bonus reward drawn as a proportion of the group reward. The group reward decreased with each new round of the game played, starting from £12.00 in total and decreasing by £0.25 per round. If no full consensus was reached after 24 rounds, then the group reward was lost and only the base reward was awarded to each participant. In the presence of recruits that dropped out of the experiment, the group reward was re-scaled proportionally to reflect the fewer number of human participants there were in the game. If a consensus was reached, then a participant's share from the group reward is proportional to the number of rounds over the course of the game in which that player chose the winning strategy (that is, the final consensus strategy), relative to the number of times everyone in the group chose the final consensus strategy. Supplementary Note 1 reports the group reward breakdown for one of the trials as an illustrative example. According to this incentive scheme, each participant's reward is maximised if he or she coordinated with others in the group, and also reached the final consensus strategy as quickly as possible. Thus, the reward structure aims to capture salient features of real-world social convention formation and evolution. For instance, the requirement for consensus to receive the group reward, in addition to its continuous decay, reflects an incentive to coordinate within a social group and the benefits of forming a convention quickly. Even so, we observed several trials in which the committed minority failed to overturn the status quo and no consensus was reached. Each participant also has an incentive to be consistent with his or her choices from previous rounds; converting others to adopt the participant's strategy to make it emerge as the winning strategy maximises the participant's share of the group reward. In the real-world, early adopters in social diffusion are often rewarded through status, experience, etc. However, if his or her current strategy does not end up being the winning strategy, then the more the participant waits, the lower the reward when the participant finally switches. From this reward structure, it is therefore plausible that both trend-seeking and inertia are observed in each participant's decision-making process, and he or she may be simultaneously influenced by both. The same monetary value was given to either strategy being the winning strategy in order to avoid any bias, to reflect the fact that in many scenarios concerning social conventions there is no clear advantage of the alternative over the status quo, and to ensure the diffusion is primarily driven by the committed minority bots. Note that, since the committed minority was fully consistent in their backing of the alternative strategy after the end of Stage I, then the alternative was the only possible winning strategy.

The ethical approval for the given study and the data management plan were authorised by the University of Groningen Faculty of Economics and Business Institutional Review Board, with reciprocal ethical approval granted by Curtin University's Human Research Ethics Committee. The participants were informed that a certain level of deception would occur in the experiments, but no specific

information, including on the presence of committed minority bots, was provided beforehand. Screenshots from the game interface are shown in Supplementary Fig. 1. Supplementary Note 1 also includes additional notes on the analysis of the experimental results.

**Parametrisation of the model**. The data used in the parametrisation process are the switching rates $y_v$ recorded from the experiments. Our parametrisation process yields the model parameters for Equation (1) and Equation (2a) and (2b), specifically $\beta$, $k_v$, $b_v$, and $r_v$.

The parametrisation consisted of three steps. First, we performed a data cleaning process, in which we identify a set of participants that had irregular behaviour. Consequently, we removed the trials in which there were too many irregular participants from the data used for the parametrisation. Second, we classified participants into two classes of explorers and non-explorers, each one characterised by different parameters. Third, we performed the parameter identification from the empirical data, in order to identify the parameter values of the ABM for the participants from the two classes.

*Data cleaning and pre-processing*. We identified those recruits that had irregular behaviour, determined as (i) those recruits that dropped out before the game began, (ii) participants that consistently missed rounds, and (iii) totally stubborn participants. A recruit that drops out before the game begins is not considered as a participant and is replaced for the entire trial by a dropout bot, which always adopts the majority strategy. A participant that misses more than 20% of the decisions over the entire length of the experiment is labelled as a player that consistently missed rounds. It may be difficult to characterise the behaviour of such a player using his or her switching rate data. We classified a participant as totally stubborn if (i) he or she plays the status quo strategy in all the rounds of the experiment and also (ii) he or she plays the status quo strategy for two consecutive rounds, when all the other players (participants and pre-programmed bots) are playing the alternative strategy. Note that an individual that plays the status quo in all the 24 rounds is not necessarily classified as totally stubborn, as we also require that the individual consistently selects the status quo even when the alternative has diffused among the rest of the group. There is no need to identify the model parameters of a stubborn player; a player $v$ will stubbornly select strategy 0 if we set $k_v = 1$ and $x_v(0) = 0$. All the 138 recruits that are not in these three categories are denoted as regular participants. Note that for participants who missed responding in 20% or fewer of the trial rounds, we treated them as a regular participant. Supplementary Note 1 contains further remarks on these three classes of recruits.

We defined in the paragraph above three types of irregularities: (i) dropouts, (ii) missing rounds, and iii) totally stubborn. We removed those trials which had at least one participant that missed rounds (that is, who missed more than 20% of the rounds) and also had more than 33% of the recruits displaying irregular behaviour. A total of 4 trials out of 20 met this criteria, and were removed for the parametrisation. After the data cleaning process, the parametrisation data consists of 119 regular participants gathered in 16 trials.

We conducted a robustness check by running the parametrisation with the data from the full set of 20 trials, i.e. with the removed trials put back into the parametrisation data set. The obtained parameters and features of diffusion identified in the simulations were robust to the exclusion of these 4 irregular trials, as reported in Supplementary Fig. 4.

Finally, we made a simple adjustment to the definition of $T^*$ and $y_v$. We first remark that $T^*$ and $y_v$ are a well-defined quantity (with probability 1) in the mathematical model, since the stochastic process is such that every strategy configuration of the population occurs with non-zero probability. However, the presence of irregular participants and the finite number of rounds being a practical limitation of the experimental setting may result in trials in which full diffusion is not observed and thus $T^*$ is not defined (as we indeed observed in 6 trials). To deal with irregular players, we define $T^*$ as the round in which full consensus of regular players is reached. For the trials in which diffusion is not observed, we define $T^* = 24$ and

$$y_v = \frac{1}{24}\left(\sum_{t=1}^{24} |x_v(t) - x_v(t-1)|\right). \quad (5)$$

Note that the $-1$ term has been removed when compared to Equation (4d). This is because the $-1$ term was used to discount the final switch of agent $v$ to the all-1 strategy configuration, since $y_v$ aims to record the switching behaviour of agent $v$ up to, but not including, the point of full diffusion. If a final switch does not occur, that discount term is not needed.

*Classification of explorers and non-explorers*. To reflect the heterogeneity observed in the experiment and keep the model as simple as possible, we defined a quantitative procedure to split the regular participants into two classes: explorers and non-explorers. The former are players that typically switch strategies several times, not necessarily toward the strategy played by the majority of the population. The latter, instead, mostly tend to switch strategy from the status quo only when there is an established majority that is playing the alternative strategy.

Based on these observations, we defined the following classification procedure. For each regular participant $v$, we counted the total number of occurrences of the following three events, before the regular participants reached a full consensus on

the alternative (or, if no consensus is reached, until the termination of the game after 24 rounds):

- the number of times participant $v$ switched strategy to the one that was being currently played by the minority of his or her fellow players (including committed minority bots), is denoted by $\text{min}_v$;
- the number of times participant $v$ switched strategy to the one that was being currently played by the majority of his or her fellow players (including committed minority bots), is denoted by $\text{maj}_v$; and
- the total number of times participant $v$ switched strategy, less 1 if the group eventually reached the all-1 consensus configuration (similar to the definition of $y_v$), is denoted by $\text{sw}_v$.

Then, we defined the behavioural discriminant of participant $v$ as the following weighted combination of the three quantities defined above:

$$\Delta_v = \frac{1}{T^*}\text{sw}_v + \text{min}_v - \frac{1}{2}\text{maj}_v, \quad (6)$$

where $T^* = 24$ if full consensus of the regular participants is not reached. We introduced the following classification rules: $v$ is a non-explorer if $\text{min}_v = 0$, or if $\text{min}_v > 0$ and $\Delta_v \leq 0$. Otherwise, if $\text{min}_v > 0$ and $\Delta_v > 0$, then $v$ is an explorer. An elucidating example is presented in Supplementary Note 3 and Supplementary Table 2.

The value $\text{sw}_v$ captures the total number of switches in strategy individual $v$ made before a full consensus was reached (or the game finished). Then, the term $\text{sw}_v/T^*$ in Equation (6) is equivalent to the switching rate $y_v$ as defined in Equation (4d). Those individuals who have a higher $\text{sw}_v$ switch strategies more often during the experiment, indicating explorer-like behaviour, although $\text{sw}_v$ does not consider whether the individual is switching to join the majority or minority strategy at that particular switch. A larger $\text{sw}_v$ contributes a larger value to the right of Equation (6), making it more likely for individual $v$ to be identified as an explorer. In Equation (6), we gave a higher weight to the switches toward the minority $\text{min}_v$ than the ones toward the majority $\text{maj}_v$. The reason for such a choice lies in the fact that it is widely accepted in the social psychology literature that deviating from an established majority to join a minority is more costly and requires more effort than the opposite change, that is, to conform with the majority[39].

Our behavioural discriminant, therefore, considers how often an individual changed strategies, whether the individual selected a minority strategy, and how often the individual joined the majority strategy.

*Parameter identification*. Our parametrisation process estimates the parameters $\beta$, $b_v$, $k_v$, $r_v$ of Equation (1) and Equation (2a) and (2b) that provide the best fit for the switching rates of the 119 regular participants identified from the 16 trials. We assumed that all participants had the same level of rationality $\beta$, and separately that all explorers and all non-explorers had the same parameters, captured by the parameters sets $b_e$, $k_e$, $r_e$ and $b_f$, $k_f$, $r_f$, respectively. Being the three parameters $b_v$, $k_v$ and $r_v$ a convex combination, and with the addition of the rationality $\beta$, this resulted in having 5 independent parameters to be estimated. Given the stochastic nature of the model, its parameters were estimated through the following Monte Carlo-based technique.

For convenience, let us define the set of the 16 trials used for the parametrisation as $\mathcal{T}$. Using the classification method above, we divided the 119 regular participants of these 16 trials into two disjoint sets of explorers, $\mathcal{E}$, and non-explorers, $\mathcal{N}$ (our identification process above led to 74 and 45 participants in $\mathcal{E}$ and $\mathcal{N}$, respectively). We then computed the mean and standard deviation of $y_v \in \mathcal{E}$ and $y_j \in \mathcal{N}$, separately, and which we denote with $\mu_\mathcal{E}$, $\sigma_\mathcal{E}$, and $\mu_\mathcal{N}$, $\sigma_\mathcal{N}$, respectively.

For each experimental trial, we created a simulation scenario that precisely reflected the trial setup. Note that among the 16 trials used in the parametrisation process, there are still irregular participants, such as totally stubborn players or players who consistently missed rounds. However, for any given trial among the 16, there are fewer than 33% of the participants displaying such irregularities. The strategy responses of these irregular participants were included in the simulation setup, as they help to shape the responses of the regular participants. However, we do not attempt to identify model parameters for the irregular participants (for reasons discussed in the Data cleaning and pre-processing subsection of the 'Methods').

To create the simulation scenario for a particular experimental trial, we first recreated the number of bots (committed minority and dropouts), and the presence (if any) of a totally stubborn participant. The 2 participants that in the 16 trials consistently missed rounds (i.e. missed more than 20% of the rounds) were considered as dropouts and substituted by two dropout bots. Because committed minority bots, dropout bots and totally stubborn participants act in a deterministic manner, we hardcoded their strategy response at each time-step into the simulation scenario rather than assigning model parameters and dynamically simulating their behaviour. Such a choice allowed us to reduce the computational effort and avoid possible numerical computational issues. For the regular participants, we set the initial strategy choice and the number of explorers and non-explorers to match the particular experimental trial.

Let $\hat{y}_v^k$ denote the switching rate of player $v$ in the simulation scenario replicating trial $k \in \mathcal{T}$. We computed 1000 independent Monte Carlo simulations

for each of the 16 simulation scenarios, and use $\hat{\mu}_{\mathcal{E}}$, $\hat{\sigma}_{\mathcal{E}}$, and $\hat{\mu}_{\mathcal{N}}$, $\hat{\sigma}_{\mathcal{N}}$ to denote the mean and standard deviation of all switching rates of explorers and non-explorers in all the trials, respectively. That is, of $\hat{y}_v^k$ for $v \in \mathcal{E}$ and $\hat{y}_w^k$ for $w \in \mathcal{N}$, respectively.

We define

$$C = a_1|\mu_{\mathcal{E}} - \hat{\mu}_{\mathcal{E}}| + a_2|\mu_{\mathcal{N}} - \hat{\mu}_{\mathcal{N}}| + a_3|\sigma_{\mathcal{E}} - \hat{\sigma}_{\mathcal{E}}| + a_4|\sigma_{\mathcal{N}} - \hat{\sigma}_{\mathcal{N}}| \quad (7)$$

to be the cost function of the parametrisation, with $a_1, \ldots, a_4$ being positive weighting coefficients. Since $b_v + k_v + r_v = 1$ for all $v$, we have in effect five independent variables to parametrise: $\beta$, $k_e$, $r_e$, $k_f$, and $r_f$. We discretised the 5-dimensional parameter space onto a regular lattice with step size of 0.2 for the rationality parameter $\beta$ and 0.01 for the other parameters. Note that the step size is adjusted according to the range of the parameter space: $\beta$ is a non-negative parameter whose value can reasonably span from 0 (randomised decisions) to 10 (high rationality), the other parameters are a convex combination, so they are bounded between 0 and 1. The model parameters were obtained as

$$\{\beta, k_e, r_e, k_f, r_f\} = \mathrm{argmin}_{\beta, k_e, r_e, k_f, r_f} C. \quad (8)$$

We selected the weighting coefficients to be $a_1 = 1$, $a_2 = 0.5$, $a_3 = 1$, and $a_4 = 1.5$ so that $a_1 + a_3 = a_2 + a_4$. This put equal weight on the switching behaviour of explorers and non-explorers. We placed less weight on the error in the mean of the non-explorers, $|\mu_{\mathcal{N}} - \hat{\mu}_{\mathcal{N}}|$, and more weight on the error in the standard deviation of the non-explorers $|\sigma_{\mathcal{N}} - \hat{\sigma}_{\mathcal{N}}|$. This was because $\mu_{\mathcal{N}} \approx 0$ and did not provide as much information about the switching behaviour of non-explorers as compared with $\sigma_{\mathcal{N}}$.

The results of our parametrisation are: $\beta = 7.8$, $b_e = 0.48$, $k_e = 0.10$, $r_e = 0.42$, $b_f = 0.42$, $k_f = 0.42$ and $r_f = 0.16$. The functional $C$ may have multiple minimum points; the value of the functional $C$ that corresponds to the best fit for the parameters is 28% less than the second minimum. The parameters that correspond to the second minimum of $C$ are reported in Supplementary Table 3 and used therein for robustness checks.

### Aspects of the simulation process

*Using $\bar{T}$ and $\Delta T$ to characterise the diffusion.* Due to the setup of our simulation, the population can remain for a period of time in a meta-stable state in which the majority of the individuals are selecting the status quo strategy. In this meta-stable state, there can be small fluctuations due to the stochastic nature of the noisy best-response dynamics, see e.g. Fig. 2a for $r_e = 0.2$ from $t = 0$ to $t = 100$. At some point in time, the alternative has diffused to a substantial (but non-majority) fraction of individuals, and the population exits the meta-stable state, and diffusion takes off.

Our work is interested in quantifying the time the population remains in such a meta-stable state, and the duration of the transition from exiting this state (the diffusion process takes off) to a pervasive adoption of the alternative. Supplementary Fig. 8 suggests that such a takeoff occurs when the diffusion process reaches a milestone in which approximately a fraction 0.4 of the entire population (including the committed minority) adopts the alternative. Hence, $\bar{T}$ defined in Equation (4b) provides a quantitative value for time required before the diffusion process takes off, while $\Delta T$ indicates the duration of the transition; the nature of the diffusion process is then indicated by the pair $(\bar{T}, \Delta T)$. If $\Delta T$ is large compared to the total duration of the diffusion process $T^*$, then the diffusion is not explosive, but rather occurs over a long period. If $\Delta T$ is small, then the diffusion process after take-off is explosive.

In fact, the take-off time $\bar{T}$ is defined as the supremum time step such that the fraction of the population (including the committed minority) adopting the alternative strategy, $\frac{1}{n}\sum_{v \in \mathcal{V}} x_v(t)$, is less than or equal to the threshold set at 0.4, before the diffusion time. Prior to $\bar{T}$, there may be fluctuations such that $\frac{1}{n}\sum_{v \in \mathcal{V}} x_v(t)$ exceeds 0.4, but will then decrease below 0.4. After $\bar{T}$, $\frac{1}{n}\sum_{v \in \mathcal{V}} x_v(t)$ never decreases below 0.4, and eventually increases until $\frac{1}{n}\sum_{v \in \mathcal{V}} x_v(T^*) = 0.99$ at time $T^*$. This ensures that $\bar{T}$ is capturing when the diffusion process takes off and subsequently does not stop, which also appears to be related to the concept of a tipping point[52].

In Supplementary Figs. 9–11 and Supplemental Note 4, we provide a robustness check by running our simulations with threshold values of 0.35 and 0.45 for defining $\bar{T}$ (changing the threshold value changes when the takeoff occurs quantitatively defined). However, the diffusion properties are defined by the pair $(\bar{T}, \Delta T)$, and these additional simulations verify that the general conclusions concerning the diffusion phenomenon, viz. the consistent presence of explosive diffusion and a delay that depends on the fraction of explorers and committed minority, are unchanged for different threshold values.

*Simulation setup.* All the numerical simulations are run using MATLAB. Monte Carlo simulations are averaged over 200 independent runs. Supplementary Note 8 provides an outline of the algorithm used for the agent-based model.

### Data availability
The data generated in this study have been deposited in the Zenodo database, under the accession code https://doi.org/10.5281/zenodo.5175151. This repository contains the raw data from the experiments (the strategy choices at each round for each individual, for every trial) but anonymised by stripping the player ID, the outputs and data of the

parametrisation procedure, and the raw data of the figures in the main article and the Supplementary Figures[46].

### Code availability
All of the code used in the simulations have been deposited in the Zenodo database, under the accession code https://doi.org/10.5281/zenodo.5175151 using an MIT License for open-source availability[46].

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

## Acknowledgements
The work of M.Y., L.Z. and M.C. was supported in part by the European Research Council (ERC-CoG-771687) and the Netherlands Organisation for Scientific Research (NWO-vidi-14134). M.Y. is also supported by the Western Australian Government, through the Premier's Science Fellowship Program.

## Author contributions
M.Y. and L.Z. proposed the model and wrote the paper and SI, with input from all authors. Z.M., J.W.B., H.R., B.M.F. and M.Y. designed the experiment. M.Y., L.Z., Z.M., J.W.B., H.R., B.M.F. and M.C. contributed equally in designing and interpreting the research; M.Y., L.Z. and Z.M. performed the research.

## Competing interests
The authors declare no competing interests.
