## [Peer Review File · Nature Communications]

Reviewers' Comments:

Reviewer #1:

Remarks to the Author:

This paper addresses a very interesting question of how innovations spread and uses two methods (experiments and modeling). I have a number of suggestions below to further enhance its impact.

1. In both the experiment and the modeling, the bots or agents play stubbornly throughout the game (i.e., they never switch their strategy despite resistance from the group). This assumption seems to be unrealistic—how often is it that innovators always stick to their minority strategy despite resistance from their groups? They would presumably be dissuaded from going the road less traveled in some circumstances. More generally, while there may be some explorers who always continue despite peer pressure, to assume that all explorers adopt this strategy doesn't seem realistic. How would relaxing that assumption change the conclusions of your model?

2. Where was the empirical data collected? I'm presuming that it was in a Western culture? Does that account for the large discrepancy of innovators? Presumably in contexts where there are strong conformity values you wouldn't find these results.

3. There seems to be a disconnect between the game the participants and agents are playing and the context to which you are generalizing—i.e. innovation diffusion framework. In both the experiments and the decision-making models, you operationalized the game as a pure coordination process. Yet in innovation diffusion, one important motivation of individuals is to figure out, through environmental and social learning, which behavior per se is a better one (see for example Henrich, 2001; Young, 2009). In many of these models, the two behaviors are not equivalent in terms of their absolute payoff and individuals need to learn which one to adopt from observing other's behavior and payoffs. However, in the current paper, it doesn't seem like this motivation was included. Instead, in the experiment, the participants' goal is solely to reach an agreement as soon as possible. Indeed, participants don't need to worry which behavior is a better one. Similarly, in the decision-making model, in Eq. (2a) and Eq. (2b), the two behaviors are equivalent. Players' payoffs depend solely on inertia and coordination but not on what behavior they actually do. This model seems focus more on a consensus rather than innovation process. Admittedly, reaching an agreement quickly is a nonnegligible goal in real-world decision-making. However, the authors may need to discuss how their study fits in or differs from the framework of innovation diffusion if the innovation itself is not related to any extra benefit.

4. You define switching rate as a key microscopic feature. I wonder what the psychological process behind it is. Especially, what's the motivation of being stubborn and what's the motivation of switching for multiple times? Under what circumstance do they or do they not switch? Though the switching rates are calculated in a similar way for both the experiment and the decision-making model, the authors may need to discuss whether switching behaviors are driven by the same mechanism/motivation in the experiment vs. in the model. In the model, players passively response to the change of strategies in the population (with some inertia). The switching processes are driven by the differences in payoffs as well as random processes in the simulation. In the experiment, however, the players may want to actively influence other's behavior or test another's preference. It is not necessarily feasible to incorporate these processes in the model, but it would be useful if you can provide more discussion on how switching behavior is related to inertia and trend-seeking, especially for the empirical data.

5. You may want to take a look at Henrich (2001) on innovation diffusion. With different assumptions and model set-ups, they also tried to explain the different shapes of diffusion curves (e.g., transition time, explosiveness, see also De et al., 2018).

6. Page 27, you note that "In stage one, at the beginning of the game, all but one innovator chose the majority supported product to help drive towards an initial consensus among the regular players, to stimulate a natural forming status quo. Stage two began once initial consensus between participants was reached, and all innovators then chose the innovation strategy, different from the status quo strategy and stubbornly continued to do so until the end of the game." How did the participants interpret the sudden change of the innovators particularly since the goal was

to reach consensus (which they did). Do you have any suspicion checks?

7. There might be a typo in Eq. (3). It would make more sense if $(n-1)$ is under the second term in the bracket. Thus, $X_v(t) = 1/2$ when the trend is neutral. Otherwise, it would be appreciated if the authors could show mathematical proof of the equation on Line 176, " $X_v(t) > 1 - X_v(t)$."

8. In Eq. (4b), what's the rationale of choosing 0.4 as the threshold? In addition, it would make more sense if tipping point is defined in terms of the rate of change of a strategy instead of the proportion of a strategy.

Henrich, J. (2001). Cultural transmission and the diffusion of innovations: Adoption dynamics indicate that biased cultural transmission is the predominate force in behavioral change. *American Anthropologist*, 103(4), 992-1013.

Young, H. P. (2009). Innovation diffusion in heterogeneous populations: Contagion, social influence, and social learning. *American economic review*, 99(5), 1899-1924.

De, S., Nau, D. S., Pan, X., & Gelfand, M. J. (2018, July). Tipping points for norm change in human cultures. In *International Conference on Social Computing, Behavioral-Cultural Modeling and Prediction and Behavior Representation in Modeling and Simulation* (pp. 61-69). Springer, Cham.

Reviewer #2:

Remarks to the Author:

Thank you for giving me the opportunity to review your paper! I am especially grateful for your inclusion of your github link and code, as it made it very straightforward for me to go deeper into your model and analysis.

Top line:

I like the way you identify behavioral phenomena (inertia and trend-seeking) that may influence social contagion, and go about exploring the implications of those phenomena in simulation and experiment. However, I'm not convinced that what you observe in your experiment, and what you describe in the simulation is fundamentally different from the Bass model. The stylistic features you identify (explosive diffusion and delayed takeoff) are also features of that model. This isn't to say that there is no difference between your model and the Bass model, but that you may need to be more explicit about how the features you highlight are unaccounted-for in existing theory. I believe you have a contribution to make, I just think it needs a bit more exploration.

On intro:

1. pg 4, line 59, you say that there are features of real-world social diffusion that existing models can't capture, but you don't say what they are. This would be a good place to be specific.

On your experiment:

2. Your experiment description in the paper is a bit vague. I can sort of infer that participants were incentivized to be in the majority, but it would be good to have a paragraph description of what participants were doing.

3. Fig 1 - I don't really understand what I'm taking away from the frequency graph? Help me understand the takeaway from the figure.

4. I didn't see a preregistration of your experimental procedure, so it would be good to show that your observations are the right ones to make, not just the ones that show up as interesting in your data. You don't need to do much for the adoption over time plots, because they are pretty standard, but the switching rate distribution seems a bit nonstandard (at least it is to me, but I'm coming mostly from a social contagion background, not a coordination game background) and as a result may want further justification.

5. I would like to see aggregate experimental results as well as representative results, summarized in a figure. You could generate a curve for each run in your experiment, and grey out all but a few that you want to highlight. (A modification of fig 1, probably).

On your simulation:

6. I love simulation for exploring the consequences of a theory. One very powerful use is to show how one theory diverges from another. You've got a good start on this with the comparison to a pure coordination game. I think it would be helpful for you to take this deeper, and use the comparison to identify the stylized features we should expect to see if your model was true that we could not see under other models.

7. I think you probably need to make a comparison to the bass model as well as the pure coordination game, so that you can identify stylized features that are truly new to the literature.

8. In the charts you show for the simulation, (Fig 2) I'd like to see if the oscillations are a normal feature of the simulation or a product of noise. One way you can do this is to show the average value over time for a large number of simulations (maybe with a shaded area for the 90% CI) plus one representative run through the simulation.

9. I appreciate your comparison with the coordination game in fig 3, but the figure is not enough to convince me that the behavior seen in your model could not be generated by the coordination game alone. I see that it is not in the three parameterizations you show in c-h, but this isn't enough to say that it couldn't show up in some other parameterization. I don't know if it is possible, but a good way to make this argument would be to draw an envelope of the behavior of all possible runs of the coordination game, and show that the envelope of your own simulation has at least some regions outside the envelope of the pure coordination game.

Even better would be to include your experimental observations and show that they too fall outside the envelope of outcomes for the coordination game. You may need to look at some more dimensions (in addition to adoption over time and switching rate) to make that comparison work, but if you could manage it, it would really sell the idea that your model generates new behavior, and explains an observed phenomenon.

10. Your model accounts for both inertia and trend seeking. Can you show the individual effects of these two, in order to show how they interact?

11. I may be out of touch with the terminology, but I don't see how your model is "evolutionary". My understanding is that evolutionary game theory models try to show how a particular behavioral trait persists in a population, when other traits are also possible. Your model may just be a "game theoretic" model without the evolutionary part.

12. An argument you may be able to use is that the mechanisms you are considering (inertia and trend following) are sufficient to generate a particular behavior, even if they are not necessary. This is useful if you can point to an empirical context where we see the behavior of interest but can't explain it with the standard models.

I've done some exploration with the SI model (a compartmental model here, but you can implement an agent version just as easily) and made some figures you might like to see, here:

<https://gist.github.com/JamesPHoughton/e249c72580c61b353d205dbb14e1530d>

Best of luck with the process,

Reviewer #3:

Remarks to the Author:

I read with great pleasure the manuscript 'Collective patterns of diffusion are shaped by individual inertia and trend-seeking' by Ye et al. The manuscript explores with a behavioural group experiment and an agent-based model simulation the evolution of collective patterns of diffusions. It finds that two parameters that seem to well characterize collective patterns of diffusions are inertia and trend-seeking. The model characterizes the conditions under which diffusion occurs.

I believe the paper explores an important topic. Although this topic (collective patterns of diffusion) has already received a significant amount of attention in the literature, the authors are able to bring an interesting perspective, borrowing concepts and insights from psychology, evolutionary game theory as well as experimental data.

I have some major and minor concerns that I have listed in the attachment. If addressed, I would be happy to see this paper published.

REVIEWER #1

This paper addresses a very interesting question of how innovations spread and uses two methods (experiments and modeling). I have a number of suggestions below to further enhance its impact.

We thank the Reviewer for the positive comments on our work, and for the valuable and insightful suggestions, which we answer in detail below.

- 1) In both the experiment and the modeling, the bots or agents play stubbornly throughout the game (i.e., they never switch their strategy despite resistance from the group). This assumption seems to be unrealistic—how often is it that innovators always stick to their minority strategy despite resistance from their groups? They would presumably be dissuaded from going the road less traveled in some circumstances. More generally, while there may be some explorers who always continue despite peer pressure, to assume that all explorers adopt this strategy doesn't seem realistic. How would relaxing that assumption change the conclusions of your model?

We wish to provide two points in response to the Reviewer's concerns.

- a) First, we wish to clarify that “innovators” and “explorers” as described in the original manuscript are two separate types of individuals. In the revised manuscript, we have clarified this by renaming “innovators” as “committed minority” (consistent with the terminology of [1]), and the first paragraph of the Introduction now explains the two separate types of individuals. “Committed minority” are individuals who stubbornly play the alternative strategy, and in the experiments, all committed minority are bots. “Explorers” are those individuals who tend to be more willing to deviate from their current strategy; when the alternative strategy is still in the minority, this means explorers mostly stick with the status quo, and will not stick stubbornly to the novel alternative strategy. Indeed, “explorers” in our experimental data are observed to switch between the status quo strategy and the novel alternative strategy several times before the game concludes.
- b) Second, we strongly believe that including the presence of a “committed minority” is a realistic and reasonable assumption in our experiment and modelling setup, as we are interested in studying social convention change via the diffusion of an alternative option to the status quo (note that we have emphasised that the problem context is social convention, at the helpful suggestion of another Reviewer). Many social changes almost invariably begin through a committed minority who, despite all odds, consistently goes against the status quo [1]. Indeed, for *minority influence* to have a noticeable impact, the committed minority individuals need to remain as consistent as possible in their stance, viz. act stubbornly [2], [3]. Of course, in the real world, the committed minority may eventually disappear if they fail to overturn the established status quo after a period of time. This is the reason why we extensively investigated the diffusion time T^* via numerical simulations (for instance, in Fig. 4 and 5 in the manuscript). In Fig. 5 for instance, the simulation time window is 50,000 time steps; simulations in which diffusion is not observed within the simulation time window reflect scenarios in the real-world in which the committed minority disappear/die out before diffusion occurs.

- 2) Where was the empirical data collected? I'm presuming that it was in a Western culture? Does that account for the large discrepancy of innovators? Presumably in contexts where there are strong conformity values you wouldn't find these results.

The participants were recruited using Prolific Academic, all being native English speakers. Thus, the Reviewer is correct to presume the majority of participants come from a Western culture. We have clarified this in the revised manuscript, in the Methods section on pg. 29–30. As clarified above, the

“committed minority” are all computer bots (the number of bots was systematically varied by us across the trials), whereas the “explorers” are human participants identified as having a tendency to change strategies often.

The Reviewer also makes a valuable point that in contexts where there are strong conformity values, we may not find precisely the same results regarding diffusion as reported in our work, using model parameters extracted from our experimental data. Cultural differences may indeed impact the likelihood or ease of securing social change, as evidenced in the work of Mutukrishna and Schaller [4]. However, we believe that the generality and flexibility of our model allow us to capture such cultural differences and explore their impact on securing social change. For instance, different cultures or environments with stronger conformity values could be captured by increasing b_v for each individual v (which governs the strength of coordination in the decision-making process), or by decreasing the fraction of explorers, or both.

In summary, the Reviewer raises an interesting and important issue around different cultures. Parametrising the model with data from different cultures could help researchers study the effects of cultural differences on social change. We therefore believe that our model could be used to better investigate this important issue, and we discuss this in the revised manuscript, in the “Discussion” section (pg. 28).

- 3) There seems to be a disconnect between the game the participants and agents are playing and the context to which you are generalizing—i.e. innovation diffusion framework. In both the experiments and the decision-making models, you operationalized the game as a pure coordination process. Yet in innovation diffusion, one important motivation of individuals is to figure out, through environmental and social learning, which behavior per se is a better one (see for example Henrich, 2001; Young, 2009). In many of these models, the two behaviors are not equivalent in terms of their absolute payoff and individuals need to learn which one to adopt from observing other’s behavior and payoffs. However, in the current paper, it doesn’t seem like this motivation was included. Instead, in the experiment, the participants’ goal is solely to reach an agreement as soon as possible. Indeed, participants don’t need to worry which behavior is a better one. Similarly, in the decision-making model, in Eq. (2a) and Eq. (2b), the two behaviors are equivalent. Players’ payoffs depend solely on inertia and coordination but not on what behavior they actually do. This model seems focus more on a consensus rather than innovation process. Admittedly, reaching an agreement quickly is a nonnegligible goal in real-world decision-making. However, the authors may need to discuss how their study fits in or differs from the framework of innovation diffusion if the innovation itself is not related to any extra benefit.

We thank the Reviewer for the insightful comment, which relates to a similar comment by Reviewer #3 on the appropriate problem context for our work. We first remark that (Henrich, 2001) and (Young, 2009) are recorded in this letter as references [5] and [6], respectively.

We have revised the manuscript so that the problem context is now centred around social diffusion in the context of bringing about changes to *social conventions* [7], [8]. This also ensures consistency with the work of Centola *et. al.*, which was a central inspiration for our study [1]. We stress that the core of our study, the model and the data are unaffected by this clarification of the problem context.

In social conventions, the value of a convention is tied to its widespread adoption and acceptance, because individuals benefit from coordinating to select the same option; a particular option being

more advantageous is secondary to coordination [7]–[10]. For instance, consider the convention of
 greeting and two possible options of bowing and handshaking. We have clarified this in the first
 paragraph of our revised Introduction. The nature of conventions is reflected in our experimental setup
 where coordination (and consequently consensus) is a central aspect of the decision-making process.
 In contrast, in the innovation diffusion context, there is often an advantage for the innovation (that
 might be learned as the Reviewer points out), e.g. hybrid corn seed may bring a better yield than
 normal corn seed.

 In the revised “Discussion” section, we briefly discuss how our framework could be extended to capture
 the novel alternative having an additional benefit as standard in the innovation diffusion literature. This
 can be easily done by including a payoff advantage $\alpha > 0$ for strategy $x = 1$, and the first term of the
 payoff function in Eq. (2a) would then read $\sum_{w \in \mathcal{V} \setminus v} (1 + \alpha) x_w(t)$. This would also bring the model
 in line with other innovation diffusion models derived from coordination games on networks, such as
 [6], [11], [12].

- 4) You define switching rate as a key microscopic feature. I wonder what the psychological process behind
 it is. Especially, what’s the motivation of being stubborn and what’s the motivation of switching for
 multiple times? Under what circumstance do they or do they not switch? Though the switching rates
 are calculated in a similar way for both the experiment and the decision-making model, the authors
 may need to discuss whether switching behaviors are driven by the same mechanism/motivation in the
 experiment vs. in the model. In the model, players passively response to the change of strategies in the
 population (with some inertia). The switching processes are driven by the differences in payoffs as well
 as random processes in the simulation. In the experiment, however, the players may want to actively
 influence other’s behavior or test another’s preference. It is not necessarily feasible to incorporate these
 processes in the model, but it would be useful if you can provide more discussion on how switching
 behavior is related to inertia and trend-seeking, especially for the empirical data.

 Switching rate is a key microscopic feature as it identifies an individual’s activity in terms of changing
 and updating his or her strategy. The precise psychological process behind changing strategy can be
 difficult to pin down, but due to the payoff structure that we adopted, we believe there are several
 salient reasons a participant may change strategy in the experiment.

- a) There is an incentive to coordinate, since the group payoff is given if and only if a full consensus
 is reached. In our model, this is captured by the coordination term, with coefficient b_v .
 b) A participant’s share of the group reward increases if he or she chooses the winning strategy
 more often. Thus, a participant may elect not to change strategy in the hopes that his or her
 current strategy is the winning strategy; in our model, we use the inertia term with coefficient
 k_v to capture this.
 c) However, if his or her current strategy does not end up being the winning strategy, then the more
 he or she waits, the lower the reward when he or she finally switches. Moreover, there is an
 incentive to reach an agreement quickly, since the group reward decays over time. These factors
 may make a participant more sensitive to the trends of the group, captured in our model with
 the trend-seeking term with coefficient r_v .

We therefore see that for any given round, there can be a variety of motives that determine the
 participant’s strategy choice for the next round. Sometimes, the motives create tension (e.g. if the
 trend is moving away from the current strategy of the participant) while at other times, the motives
 reinforce (e.g. if the trend is growing toward the current strategy of the participant). A participant may
 elect to switch strategy to resolve any tension, which gives us motivation to examine the switching rates

of the participants. Our focus on the switching rates in the agent-based model is in part motivated by a desire to ensure the model is consistent with the data both at the group/macroscale level (diffusion curves) and at the individual level.

In the revised version of the manuscript, we have provided more details to elucidate the payoff structure for the experiment, and why this may result in inertia and trend-seeking being reflected in the empirical switching rates recorded. The changes are found in the subsection “Experimental evidence” and in the “Methods” section, on pg. 8 and 32–33, respectively.

- 5) You may want to take a look at Henrich (2001) on innovation diffusion. With different assumptions and model set-ups, they also tried to explain the different shapes of diffusion curves (e.g., transition time, explosiveness, see also De et al., 2018).

We thank the Reviewer for the useful reference suggestions, which have been included in our exploration of diffusion literature in the revised “Introduction” section. Here, we provided some further comments on the two papers of (Henrich, 2001) and (De *et. al.*, 2018), which are references [5] and [13] in this letter, respectively.

- a) Both Henrich and De *et. al.* examine population models. In our Introduction, we discuss the strengths and weaknesses of population models in general. A key issue is that it can be challenging to incorporate individual-level empirical data (such as the highly heterogeneous switching rates in our experiments) into population models. A second issue is that in social diffusion, the individual-level decisions can depend on the number and nature of interactions between an individual and his or her neighbours, as indicated by the complex contagion literature [14]. This further motivates us to consider agent-based models, which are better able to link individual-level dynamics to the emergent population-level behaviours.
- b) Henrich compares an existing model (environmental learning model) to a newly proposed model (biased transmission model), with the latter being related to discrete-time replicator equations (describing imitation dynamics, which is another classical concept in evolutionary game theory).
- c) De *et. al.* consider replicator dynamics but in continuous time and with an additional component representing social coordination/conformity. This work shows that the strong presence of conformity can hinder norm change (diffusion) initially, but actually accelerate change once a tipping point is reached.

Both papers consider models that can produce different S-shaped diffusion curves, where the transition time can be small (explosive diffusion), but the tipping point can be delayed. Crucially, both papers focus on diffusion when the novel alternative is superior to the status quo; while this is indeed observed in a variety of real-world scenarios, it is not always the case. In our revised manuscript, the common findings between the works (delayed tipping point, explosive diffusion) are considered, while our contributions are still highlighted: inertia and trend-seeking can have a significant effect in shaping diffusion patterns when there is a committed minority promoting an alternative that is not obviously superior (as may occur in social conventions).

- 6) Page 27, you note that “In stage one, at the beginning of the game, all but one innovator chose the majority supported product to help drive towards an initial consensus among the regular players, to stimulate a natural forming status quo. Stage two began once initial consensus between participants was reached, and all innovators then chose the innovation strategy, different from the status quo strategy and stubbornly continued to do so until the end of the game.” How did the participants interpret the sudden change of the innovators particularly since the goal was to reach consensus (which they did).

Do you have any suspicion checks?

Participants were only able to see the **percentage** of other participants that had selected Product *Eta*
vs. Product *Tao*, and **could not** see which other specific participant, including the committed minority
bots (termed “innovators” in the previous manuscript) had selected which product. Combined with the
synchronous updating of the decisions, a participant would only observe that a certain proportion of
the group has switched from the majority to the minority product, but not identify who switched at
which round. We have clarified this in the revised manuscript, on pg. 7 and 31.

We did not check suspicion directly, but did provide participants with an ability to leave open comments
at the end. A large number of the comments described feeling frustration that certain people were
holding off a consensus, but only very few mentioned suspicion that this minority were not real people.
This suggests the suspicion levels were not high in our samples.

- 7) There might be a typo in Eq. (3). It would make more sense if $(n - 1)$ is under the second term in
the bracket. Thus, $X_v(t) = 1/2$ when the trend is neutral. Otherwise, it would be appreciated if the
authors could show mathematical proof of the equation on Line 176, “ $X_v(t) > 1 - X_v(t)$.”

We thank the Reviewer for pointing out this typo. We have amended Eq. (3) accordingly.

- 8) In Eq. (4b), what’s the rationale of choosing 0.4 as the threshold? In addition, it would make more
sense if tipping point is defined in terms of the rate of change of a strategy instead of the proportion
of a strategy.

We thank the Reviewer for this comment, which has given us the opportunity to better clarify our
rationale behind our definition of the tipping point. First of all, we want to highlight that our work
is interested in quantifying the time taken to reach a “tipping point” (which can be delayed), and
also subsequently the duration of the transition from the tipping point to a pervasive adoption of the
alternative. These two aspects are captured by \bar{T} and ΔT , respectively, and the nature of the diffusion
process is then indicated by the pair $(\bar{T}, \Delta T)$.

Next, we note that the threshold in Eq. (4b) includes the committed minority. For instance, given a
threshold of 0.4 and a fraction of 0.25 committed minority, the tipping point requires another fraction
of 0.15 switching to the alternative (that is 20% of the remaining 75% of the population, comprised
of explorers and non-explorers). The threshold value should there be less than 0.5 (otherwise at the
threshold the alternative strategy is already the majority strategy), and sufficiently above the fraction
of committed minority such that random fluctuations of a small number of regular agents away from
status-quo majority do not exceed the threshold.

To better motivate and support our choice of setting such a threshold at 0.4, we have reported in the
Supplementary information the new Fig. S8, which shows the velocity of the diffusion process (that
is, the rate of change), as a function of the current number of adopters of the alternative. Such a plot
positions the “tipping point” of the diffusion process approximately about 0.4. In fact, below such a
threshold, the system mostly fluctuates about the meta-stable status-quo majority state (with the average
velocity reaching a minimum); once the threshold is reached, the velocity of the diffusion starts to grow.

We have also conducted a robustness check by choosing different values rather than 0.4. In particular,

the revised Supplementary Information, Figs. S9–S11 now show simulations where the threshold is
set to 0.35 and 0.45, for 0.17, 0.25 and 0.33 fraction of committed minority. One can see that the
key qualitative aspects of the diffusion patterns are unchanged (i.e. explosive diffusion is preserved,
while the growth in the delay is still dependent on the fraction of committed minority and fraction of
explorers).

Besides the new Supplementary Information, we have also revised the manuscript to better explain
how \bar{T} and ΔT are able to describe the diffusion process. This is presented in the “Methods”, in a
new section on pg. 42 titled “Using \bar{T} and ΔT to characterise the diffusion”.

REVIEWER #2

Thank you for giving me the opportunity to review your paper! I am especially grateful for your inclusion
 of your github link and code, as it made it very straightforward for me to go deeper into your model and
 analysis.

Top line: I like the way you identify behavioral phenomena (inertia and trend-seeking) that may influence
 social contagion, and go about exploring the implications of those phenomena in simulation and experiment.
 However, I'm not convinced that what you observe in your experiment, and what you describe in the
 simulation is fundamentally different from the Bass model. The stylistic features you identify (explosive
 diffusion and delayed takeoff) are also features of that model. This isn't to say that there is no difference
 between your model and the Bass model, but that you may need to be more explicit about how the features
 you highlight are unaccounted-for in existing theory. I believe you have a contribution to make, I just think
 it needs a bit more exploration.

Thank you for your positive comments. Below, we have provided detailed responses which seek
 to clarify the contributions of our work. We clarify that the Bass model is a population model, which is
 able to accurately describe population-level phenomena, but is limited in its ability to link individual-level
 mechanisms and processes to the emergence of said phenomena. In particular, our work focuses on the role
 of individual-level inertia and trend-seeking in producing the explosive diffusion and delayed take-off which
 are captured by the Bass model, and also empirical data.

On intro: 1. pg 4, line 59, you say that there are features of real-world social diffusion that existing models
 can't capture, but you don't say what they are. This would be a good place to be specific.

We have revised the Introduction to provide more clarity, noting that the relevant content is now on pg. 5,
 line 72 in the revised manuscript. In particular, we clarify that existing agent-based models cannot produce
 features such as a long delay before reaching the tipping point, followed by explosive diffusion, *without*
 *introducing agent-level assumptions that are unrealistic*. These unrealistic agent-level assumptions produce
 agent dynamics that show high switching rates (indicating frequent flip-flopping between strategies), which
 are inconsistent with the literature on inertia. Later on pg. 17–19 and the associated Fig. 3, we provide more
 details, showing the agent dynamics are also inconsistent with the empirical data we gathered.

On your experiment: 2. Your experiment description in the paper is a bit vague. I can sort of infer that
 participants were incentivized to be in the majority, but it would be good to have a paragraph description
 of what participants were doing.

We have revised the manuscript to provide more details on the experimental description. In particular

- 1) In the main **Results** section (pg. 8 of the revised paper), we now additionally describe the reward
 system, and also the general ideas around what is being incentivised.
- 2) In the **Methods** section (pg. 32–33 of the revised paper), we have provided further details on what
 the reward system incentivised, and elaborated further on the rules of the game.

3. Fig 1 - I don't really understand what I'm taking away from the frequency graph? Help me understand
 the takeaway from the figure.

We thank the Reviewer for this comment, which have allowed us to improve the discussion on Fig. 1. In
 particular, there are several important properties of the switching rate distribution which we wish the reader
 to take note of, and associated conclusions.

- • The first is the highly heterogeneous nature of the distribution, which suggests that there is a strong
heterogeneity in the way that participants approached the game, despite being offered the exact same
incentive structure.
- • The second is the shape of the distribution. There is a large peak in the distribution around 0,
containing around two thirds of all participants. The remainder of the participants have switching
activities distributed between 0.05 and 0.4.
- • This heterogeneity in the distribution provides motivation for us to consider two classes of agents (with
different parameters) in our agent-based model when conducting the parametrisation using the empirical
data. Although further heterogeneity can be introduced, there is no compelling reason to do so, as we
show that having two classes of agents is sufficient to capture the shape of the empirical distribution.
- • Finally, we see in subsequent analysis (Fig. 3 of the manuscript) that existing models without inertia and
trend-seeking are not able to reproduce the characteristics of this distribution. This provides evidence
that existing models can produce macroscopic diffusion curves with delay and explosive diffusion only
if unrealistic microscopic assumptions are adopted.

In the revised manuscript, we have expanded the caption of the figure (now, Fig. 1b), and also provided
further discussion in the Results section covering the experimental data, on pg. 10–11.

4. I didn't see a preregistration of your experimental procedure, so it would be good to show that your
observations are the right ones to make, not just the ones that show up as interesting in your data. You don't
need to do much for the adoption over time plots, because they are pretty standard, but the switching rate
distribution seems a bit nonstandard (at least it is to me, but I'm coming mostly from a social contagion
background, not a coordination game background) and as a result may want further justification.

We did not complete pre-registration for our experimental procedure. The focus on our work was in explor-
ing modelling; the experiment first and foremost provided us with usable individual-level data to parametrise
the agent-based model, and identified salient microscopic-level phenomena that required consideration (e.g.
heterogeneous switching activity).

Nonetheless, we recognise the importance of transparency and reproducibility in empirical studies. We
have uploaded all of the raw data from the experiments to <https://github.com/lzino90/diffusion>; this contains
the strategy choice for each individual (participants and computer bots) at each round, over all experimental
trials. We have elected to focus on the presence of inertia and trend-seeking (especially a heterogeneous
presence) at the individual-level and how this translates to modelling methodology; this does not preclude
identification of other interesting observations from our data which may be suitable for further analysis, and
all data are provided to ensure this can be done with full transparency.

With regards as to why we consider the use of switching rates, this is because switching rates provide us
an understanding of individual-level dynamics, in particular how often each player decides to change his or
her current strategy. When this is paired with the adoption over time, we are able to obtain an understanding
of when and how often players change their strategies. Our experiment presents a reward structure that
has several different incentives (which can sometimes clash or reinforce), and a player must balance and
reconcile them to determine if he or she should switch strategy; the switching rate distribution thus provides
358 us with an intuitive understanding of how the individuals are impacted by the different incentives, including
359 heterogeneity between the individuals (see our response above, to Reviewer #1, Point 4, for details on these
360 incentives).

5. I would like to see aggregate experimental results as well as representative results, summarized in a
figure. You could generate a curve for each run in your experiment, and grey out all but a few that you

want to highlight. (A modification of fig 1, probably).

Thank you for the helpful suggestion. We have now revised Fig. 1a to show all experimental trials, but
 in light grey lines to aid clarity. We have plotted 3 trials with bold coloured lines, giving examples of
 rapid diffusion (green), delayed explosive diffusion (blue), and no diffusion (red). These 3 trials were the
 trials shown in the original manuscript. Each trial is also independently presented in the Supplementary
 Information, Fig. S2.

On your simulation: 6. I love simulation for exploring the consequences of a theory. One very powerful
 use is to show how one theory diverges from another. You've got a good start on this with the comparison
 to a pure coordination game. I think it would be helpful for you to take this deeper, and use the comparison
 to identify the stylized features we should expect to see if your model was true that we could not see under
 other models.

We are grateful that you have appreciated our simulation-based approach and we thank you for the
 suggestions. From our simulations, we expect to *simultaneously* observe the stylized features of i) explosive
 diffusion, ii) a potential for delayed take-off, and iii) individual-level dynamics that reflect data (moderate
 and heterogeneous switching rate with a peak close to 0). We have clarified this in the revised "Introduction"
 section. Further to this, we expanded our analysis of the coordination game, and for the model with only
 inertia or only trend-seeking (parametrised using our empirical data). This is reported below in Points 9 and
 10, respectively, and in the revised manuscript. Our findings show that the three stylized features cannot
 simultaneously appear unless trend-seeking and inertia are both incorporated into the model.

7. I think you probably need to make a comparison to the Bass model as well as the pure coordination
 game, so that you can identify stylized features that are truly new to the literature.

The Bass model is a population model, which takes a macroscopic approach to describing social diffusion
 phenomenon. In our revised manuscript (primarily in the Introduction and Discussions sections, on pg. 3
 and 27, respectively), we discuss the strengths and weaknesses of population models in general (including
 the Bass model, and several identified above by Reviewer #1). We explain that we adopted an agent-based
 model instead, which takes a microscopic approach, because this allows us to explore how individual-level
 mechanisms of inertia and trend-seeking shape the emergent diffusion phenomena. We summarise the key
 points here:

- 1) Population models can produce S-curves with different characteristic features such as delayed take-off
 and/or explosive diffusion. This can be achieved with a minimal number of parameters, which is a clear
 strength of population models. We are not claiming that the Bass model (or other population models)
 cannot reproduce the macroscopic features observed in our model. However, population models have
 certain drawbacks.
- 2) With population models, it is not straightforward to understand how individual-level mechanisms
 (especially if there are several as in our work) can critically shape the emergent macroscopic behaviour.
- 3) It can be difficult to incorporate data gathered at the individual-level (such as the switching rate
 information from our experiments) into population models.
- 4) Agent-based models can be more easily generalised to incorporate network structure effects, individual-
 level interventions or targeted incentive policies, or population heterogeneity. Our work identified the
 clear heterogeneity in the population, and showed the impact this can have in the diffusion process.
 The other aspects are promising directions for future work, which can be explored by extending our
 basic modelling framework. For instance, although we assumed an all-to-all interconnection in our
 model (ensuring consistency with our experimental setup, and allowing us to focus on the study of

inertia and trend-seeking), it is trivial to extend our model to include a network structure.

8. In the charts you show for the simulation, (Fig 2) I'd like to see if the oscillations are a normal feature
 of the simulation or a product of noise. One way you can do this is to show the average value over time
 for a large number of simulations (maybe with a shaded area for the 90% CI) plus one representative run
 through the simulation.

Following your suggestion, we have expanded Fig. 2 by additionally showing the diffusion curves for
 50 simulations for each given ratio of explorers, ρ_e , with faint transparent lines. For each given ρ_e , a bold
 line indicates a representative simulation example. Note that presenting quantities averaged over all the
 simulations may obscure the characteristic features we would like to show, such as the explosiveness of
 the transition. Our stochastic model uses a noisy best-response dynamics to capture bounded rationality in
 the decision-making process, which means each individual has a positive probability for selecting either the
 status quo or the alternative, and is the cause of the oscillations; we believe that our revised figure now
 clarifies this.

9. I appreciate your comparison with the coordination game in fig 3, but the figure is not enough to
 convince me that the behavior seen in your model could not be generated by the coordination game alone. I
 see that it is not in the three parameterizations you show in c-h, but this isn't enough to say that it couldn't
 show up in some other parameterization. I don't know if it is possible, but a good way to make this argument
 would be to draw an envelope of the behavior of all possible runs of the coordination game, and show that
 the envelope of your own simulation has at least some regions outside the envelope of the pure coordination
 game.

Even better would be to include your experimental observations and show that they too fall outside the
 envelope of outcomes for the coordination game. You may need to look at some more dimensions (in addition
 to adoption over time and switching rate) to make that comparison work, but if you could manage it, it
 would really sell the idea that your model generates new behavior, and explains an observed phenomenon.

Thank you for giving us this opportunity to expand on the important comparison between our model and
 the possible outcomes of the coordination game. In the revised manuscript, we have provided further analysis
 of the outcomes of the pure coordination game, driven by new parameter identification and simulations. We
 believe these new findings strengthen our paper, providing additional supporting evidence that the pure
 coordination game cannot capture our experimental observations.

We first provide a brief remark on the full envelope of coordination game outcomes. In a model based
 on a pure coordination game (and in which both strategies yield the same payoff) there are n degrees of
 freedom to parametrise the model, being the rationality β_i for the n individuals in the population. Thus, it
 is conceivable that by selecting a specific distribution of β_i , and given a specific initialisation of the model,
 one can obtain the diffusion behaviour observed from our model (e.g. Fig. 3a and 3b in the main paper).
 However, this risks over-fitting the model to the data (given the large n degrees of parameter freedom), and
 goes against the strengths of agent-based models: identifying how individual-level dynamics governed by
 intuitive rules can lead to robust predictions on complex population-level phenomena.

Our expanded analysis begins by parametrising a model with just coordination (obtained by setting our
 model to have no inertia or trend-seeking) using the same parametrisation technique detailed in our work,
 and using the same empirical data we collected. Since $b_v = 1$ and $k_v = r_v = 0$ in a pure coordination
 setting, we allow the rationality to differ between the explorers and non-explorers, β_e and β_f , to capture
 heterogeneity while ensuring the degree of freedom remains reasonable. The new findings are reported in
 the revised manuscript, presented in the new Fig. 3 and discussed on pg. 17–19. We summarise the key

findings here:

- 1) Simulations with the best-fit parameters with only coordination are shown in Figs. 3c and 3d (new
 figures). First, we notice that there was a higher cost of the functional used in the parametrisation
 (which accounts for the error between model and data) compared to our proposed model (specifically,
 it increased by 8140%), which suggests that having only the coordination mechanism is not sufficient
 to reproduce the empirical data. More importantly, diffusion was not observed in any of the 50
 simulation runs, which we attribute to the high rationality of the non-explorers preventing a tipping
 point from being reached within our simulation time window of 100,000 rounds. The individual-level
 switching rates show heterogeneity, but the shape of the distribution between the simulations and the
 empirical data are not the same. In particular, Fig. 3d shows a bimodal distribution with two sharp and
 concentrated modes at 0 and 0.225, respectively, which contrasts our empirical data that shows a large
 peak at 0 and then a wide distribution of switching rates between 0.1 and 0.4. Thus, the coordination
 game, parametrised using our empirical data, can to a certain degree capture the microscopic features
 of the diffusion behaviour, but cannot capture the macroscopic features.
- 2) Based on the parameters of rationality β obtained for the coordination only model, it is possible to
 generate delayed and explosive diffusion at the macroscopic level by either i) increasing the fraction of
 explorers or ii) reducing the rationality β_f for the followers. This is shown in the revised manuscript,
 Figs. 3e–h (new figures). While this adjustment allows the coordination model to capture macroscopic
 outcomes similar to our model, one can see that the microscopic level observations (Figs. 3f and 3h)
 do not reflect the empirical data. In particular, if the number of explorers is increased, Fig. 3f shows a
 very high switching activity (more than half of the agents switch at least once every 4 rounds), with the
 largest peak (most of the players) at about 0.3; if the rationality of non-explorers is increased, Fig. 3h
 shows that the switching rates tend to be quite homogeneous and high, with a large peak concentrated
 around $y \approx 0.2$ (switching once every 5 rounds). None of the plots are consistent with Figs. 3b and
 with our empirical data (Fig. 1b), which show a large peak close to 0 and a very heterogeneous
 switching rate distribution.

When compared to the model that includes inertia and trend-seeking (Fig. 3a and 3b), we believe the
 additional analysis now reported in the manuscript gives evidence that our experimental observations fall
 outside the envelope of outcomes for a coordination game model. We achieved this by parametrising the
 coordination game model using our data (at the switching rate level), allowing for some heterogeneity in the
 rationality parameter. As the coordination game could not simultaneously capture our switching rate data
 (microscopic level) and delayed and explosive diffusion (macroscopic level), we did not pursue analysis of
 additional dimensions.

10. Your model accounts for both inertia and trend seeking. Can you show the individual effects of these
 two, in order to show how they interact?

Thank you for the interesting question. As suggested, we explored the individual effects of these two
 mechanisms, and found they interact in a complex and non-negligible manner. The details are presented
 in full in the Supplementary Information, Section S5, and briefly discussed in the revised manuscript on
 487 pg. 17–19. We summarise the findings here for your convenience.

To begin, we ran the parametrisation process (as described in the Methods section) by forcing either
 $r_v = 0$ or $k_v = 0$, corresponding to no trend-seeking or no inertia, respectively. The parameters obtained
 are reported in the Supplementary Information, Table S4. We then ran simulations with an identical setup
 as that appearing in Figure 2 in the main article, but with the new parameters.

1) The results for when there is only trend-seeking appear in Fig. S14. At the microscopic level, the

switching rate data shows heterogeneity similar to the empirical data, but at the macroscopic level, we observe that diffusion does not occur. This is due to the large rationality parameter $\beta = 14$, which prevents non-explorers from switching often.

- 2) The results for when there is only inertia appear in Fig. S15. At the microscopic level, the switching rate data shows heterogeneity similar to the empirical data. At the macroscopic level, we observe that diffusion always occurs, but without delay. Reducing the fraction of explorers, ρ_e , increases the diffusion time T^* but also flattens the diffusion curve, so that the diffusion is never explosive. This is similar to what is obtained from the SI model as illustrated in the GitHub link you kindly provided. Diffusion occurring when there is inertia appears counter-intuitive at first, but is reasonable once it is noticed that there is a very low rationality parameter $\beta = 5.8$ (as identified by the parametrisation process).

To summarise then, inertia and trend-seeking combine in a non-trivial way to produce diffusion phenomena that are realistic at both the microscopic and macroscopic level. It is the interplay between the two mechanisms that generates the delayed but explosive diffusion, with the strongly heterogeneous individual-level activity. Omitting either of the two mechanisms produces uninformative model predictions.

11. I may be out of touch with the terminology, but I don't see how your model is "evolutionary". My understanding is that evolutionary game theory models try to show how a particular behavioral trait persists in a population, when other traits are also possible. Your model may just be a "game theoretic" model without the evolutionary part.

While evolutionary game theory originally emerged from a biological context [15], there have been recent applications of evolutionary game theory in many different fields where "evolution" may refer to social or cultural changes, see e.g. [13], [16], diffusion of innovation, see e.g. [6], [11], or decisions on whether to take a vaccination shot during an epidemic outbreak, see, e.g. [17]. However, we also appreciate your point, and we have entirely removed the term "evolutionary" from the revised manuscript, and simply refer to our model as a "game-theoretic" model. We believe this ensures clarity, and places focus on how inertia and trend-seeking can shape the diffusion process, rather than the specifics of the game-theoretic framework we have used as the modelling basis.

12. An argument you may be able to use is that the mechanisms you are considering (inertia and trend following) are sufficient to generate a particular behavior, even if they are not necessary. This is useful if you can point to an empirical context where we see the behavior of interest but can't explain it with the standard models.

Thank you for pointing out the important distinction between sufficiency and necessity. Based on our results, including the extended analysis of the coordination game reported in response to your Point 9 above, we can provide the following conclusions.

The standard coordination game model alone cannot reproduce the phenomena of interest at both the microscopic and macroscopic level. Inertia alone, and trend-seeking alone, are not sufficient to generate the phenomena of interest. Incorporating inertia and trend-seeking *together* to extend the coordination game model is sufficient to generate the desired diffusion patterns. Of course, there may be other mechanisms one could add to the coordination game, besides inertia and trend-seeking, which generates the phenomena of interest. Thus, we cannot claim that inertia and trend-seeking are necessary to witness delayed and explosive diffusion, and moderate switching rates.

We have carefully reviewed the manuscript and removed any statements in which it was erroneously appearing that we were claiming necessity.

I've done some exploration with the SI model (a compartmental model here, but you can implement an
agent version just as easily) and made some figures you might like to see, here:

<https://gist.github.com/JamesPHoughton/e249c72580c61b353d205dbb14e1530d>

Best of luck with the process

,

Thank you for this suggestion, and for exploring further using an SI model. From your material, we
find that the SI model produces diffusion curves in which the transition time grows proportionally with the
tipping point time. That is, the larger the delay, the less explosive the transition. As a consequence, the
SI model can produce i) highly explosive transition with no (or very short) delay, ii) moderate explosive
transition with moderate delay, or iii) almost linear growth with a long delay. Hence, it seems that the SI
model cannot simultaneously produce a long delay with an explosive transition (at least without significant
modifications), as observed in our model, see Fig. 2a, and also Fig. 4.

This is also in accordance with our findings in the Supplementary Information, Section S4.1 and Fig. S13
on "epidemic-like models". We have updated the comparison with a stochastic agent-based SIS model (which
allows individuals to go back to the status quo, in contrast to the SI model), simulating a model inspired
by the one that you suggested.

REVIEWER #3

I read with great pleasure the manuscript ‘Collective patterns of diffusion are shaped by individual inertia
and trend-seeking’ by Ye et al. The manuscript explores with a behavioral group experiment and an agent-
based model simulation the evolution of collective patterns of diffusions. It finds that two parameters
that seem to well characterize collective patterns of diffusions are inertia and trend-seeking. The model
characterizes the conditions under which diffusion occurs.

I believe the paper explores an important topic. Although this topic (collective patterns of diffusion)
has already received a significant amount of attention in the literature, the authors are able to bring an
interesting perspective, borrowing concepts and insights from psychology, evolutionary game theory as well
as experimental data.

We thank the Reviewer for the positive comments on our manuscript and the contributions of our work.
Below, we have provided detailed responses to your comments and concerns, and identified what changes
have been made in the revised manuscript.

*Major Concerns*

- Ln 122. I don’t think many readers will agree with you that Figure 1d is bimodal. Perhaps representing
the distribution as a density or violin plot may avoid issues with bins definition. Still, I don’t think you
can claim it’s bimodal. This was originally a minor comment, but by reading further I noticed you put a
lot of emphasis on the (purported) bimodality of this distribution. Similarly, in Ln 277-281 you suggest a
close match between experimental and model data. I would discuss in details this match (or lack thereof)

We thank the Reviewer for pointing out this issue. In hindsight, we agree that labelling the distribution
in Fig. 1b (this was Fig. 1d in the original manuscript) as bimodal was not appropriate. In the revised
manuscript, we have removed all references of “bimodal”. Moreover, such a labelling is not necessary,
and does not have any material impact on the approach or conclusions of our work.

In the revised manuscript (pg. 10), we stress two key points to notice from the empirical switching rates:
i) the average switching rate is low with a peak around 0, and ii) there is a large degree of heterogeneity.
These two key points also motivate central elements of our modelling approach. In particular, the first
point suggested the presence of inertia, which was subsequently supported by the parametrisation process
(detailed in Supplementary Information, Fig. S14). The large degree in heterogeneity pointed to the fact
that in decision-making, and when faced with the same scenario, some people will do one thing, while
others will do something different. Our introduction of two classes of individuals into the agent-based
model simulations, viz. “explorers” and “non-explorers”, is the simplest way to introduce heterogeneity
into the agent population. We have also clarified in the manuscript what we mean by a “close match”,
see pg. 17.

- Ln 96. It is crucial for appreciating the results that you provide an expanded description of the rewards
you gave to participants to the behavioural experiment. Clearly understanding the incentives of the game
is crucial for the reader if you want them to accept your definition of what rational or irrational behaviour
is later in the paper.

In accordance with the Reviewer’s suggestions, and consistent also with comments from Reviewer #2,
we have provided further details on the rewards to the participants in our experiments. In particular

- 1) In the **Results**, in the section “Experimental evidence” on pg. 7–8, we have expanded the description of how the participants are rewarded in the experiment, providing specific details.
- 2) In the **Methods**, besides explaining how the reward system functions, we have now provided some discussion on how such a system might offer particular incentives for a participant, and what potential effects/experiences might arise. See pg. 32-33 of the revised manuscript.
- 3) In the **Supplementary Information** (pg. 7–8 and Tab. S2), we now illustrate the rewards that each participant received in one representative trial of the experiment.

- Ln 142. I would like to see better clarified what you define as *status quo* and what as innovation. From what I understand, your definition of *status quo* vs innovation is the solutions that were adopted by the majority early on vs later in the rounds. However, others may interpret the *status quo* simply as the option selected by the majority (even if this was an innovation before the majority adopted it). I was confused by your use of *status-quo/innovation* in Ln 142 because players are equally split 50-50% between solutions.

We first mention that the two strategies are now referred to as the *status quo* and the *alternative*, and we no longer explicitly reference an “innovation”. This was to provide a more nuanced view of where our model and experimental results are applicable, as suggested by yourself below and also another Reviewer. Our manuscript now proposes the work as an investigation of changes in social conventions due to social diffusion of an alternative, making it consistent with [1]. This also removes confusion around an “innovation”. In our response to your comment further below, we expand on why we believe framing our problem in the context of social conventions is appropriate.

In Ln 142 (now Ln 177 in the revised manuscript), our discussion was primarily focused on explaining how the presence of trend-seeking can be observed from the data. We have clarified the wording around this part.

Nonetheless, we also clarified the definitions of *status quo* and *alternative* in the text where we describe the experimental procedure in detail. We also clarify that the committed minority bots first act to establish a *status quo* strategy adopted by all regular participants and all but one bot (this is achieved within the first 1 – 3 rounds of the game). Then, the committed minority bots stubbornly select the *alternative* strategy to stimulate diffusion. The revisions are found in Ln 133.

- Ln 288-289 I am not convinced by your wording. Couldn’t the same sentence be stated swapping the terms ‘innovator’ and ‘explorer’ and still be correct? Please explain why you attribute the key role to innovators and the moderation role to explorers and not vice versa. Furthermore, the term moderation has a very specific connotation in statistics, which I don’t think you mean here.

We thank the Reviewer for pointing out this issue. First, we agree that the term “moderation” is not appropriate, and we have accordingly removed the term “moderation” from the revised manuscript, and then rewritten the discussion on the results of our simulations, to improve its readability.

In particular, we have clarified our reasoning for attributing the key role to the committed minority (previously called innovators), and then discuss the implications of the findings. Referring to Fig. 5, notice the following.

- 1) For any given fraction of explorers, ρ_e , there exists a threshold value for the number of committed minority, $|Z|/n$, that allows for fast diffusion (green sectors on the heat map). Indeed, if the

committed minority are more than 26% of the agent population, then we witness fast diffusion irrespective of the fraction of explorers.

- 2) The converse is not true; given a fixed number of committed minority comprising less than 26% of the population, there are values of ρ_e for which one never observes fast diffusion (white sectors on the heat map). Indeed, when less than 16% of the population are part of the committed minority, fast diffusion is never observed regardless of the number of explorers.

This finding further expands upon our understanding of diffusion and societal change. Change in society often involves a small set of people who stubbornly try to overthrow the status quo (termed zealots in some literature). Based on existing literature such as [1], we know that a tipping point will be reached and diffusion will occur when there are a sufficient number of committed minority. In our work, this is 26%, which closely aligns with the 25% identified in [1]. However, in many real-world circumstances, the committed minority are often quite marginal and may not reach the critical mass of 25%. Our findings indicate when the number of committed minority is below this critical mass, a tipping point can still be reached and diffusion can occur, if there are enough “explorers” that are willing to jump on board, similar to the concept of “early adopters” in the framework of Rogers [18].

These points have been incorporated into the revised manuscript, in the discussion of the results of Fig. 5 on pg. 24–26.

- I understand that it is tempting to mention the COVID pandemic to illustrate your coordination example, but I don’t think this fits very well with your model. Why would people be more willing to wear masks because the notice a trend towards mask-wearing (your r parameter)? Once an individual wears a mask, they are protected from contagion (assuming good masks), so their payoff (the value of wearing a mask) is independent of how many other people are wearing a mask. There is no trend-seeking there. Your model seems to be better equipped to explain technology adoption, e.g. joining a social network, where the number of people joining the social media platform correlates with your value in joining the social media platform. I think it is important to adopt a more nuanced view of what your model can and cannot explain, especially in the introduction and discussion, instead of claiming that it explains all phenomena of diffusion.

Thank you for the insightful comment. There are complex factors determining the efficacy of face masks in preventing the spread of COVID-19 (including the type of mask, and whether it prevents the wearer from spreading COVID-19 vs. protecting an uninfected wearer [19], [20]). Providing a detailed discussion around the COVID-19 pandemic would unnecessarily distract from the main findings of our work, and we have therefore removed all mention of this in the manuscript.

Nonetheless, the Reviewer rightly points out the importance of carefully considering the applicability of our work, and what it can and cannot explain. Toward that end, we have revised our work to be framed in the context of *social conventions*, with social diffusion leading to the adoption of an *alternative* over a *status quo*. There are several reasons motivating this, although we stress that this does not materially affect our experiments, the model or the results. Rather, we believe the new framing clarifies our contribution.

- 1) In our experiment, neither strategy provides an advantage in the reward. Rather, the reward structure favours coordination and consensus, early adoption and also remaining consistent (detailed explanations are provided in the revised manuscript, in the Methods section, pg. 32–33). This reward structure is adept to describe many instances of social conventions, where individuals benefit from coordinating to select the same option, rather than because of an advantage of a particular

option [7]–[10]. For instance, consider the use of different spellings of “centre” and “center” [21]. Nonetheless, players may still be sensitive to trends and the desire to remain consistent due to the social nature of the interactions.

- 2) Our model involves a coordination game in which neither strategy has an advantage in the payoff in order to remain consistent with the experimental design, and this flows through to all of the results presented in our work.
- 3) In both the experiments and the model and simulations, the diffusion process is driven by a committed minority of individuals, which can often be observed in the evolution of social conventions [22], [23]. The “social convention” framing also allows this work to align with that of [1], which was a key work that inspired our own study.
- 4) In contrast, a significant proportion of the innovation diffusion literature posits that the innovation provides a nontrivial benefit over the status quo, and this is key to the diffusion. Directly framing our work, and the reported results, as innovation diffusion would therefore yield some inconsistency and would impact the way one would compare our work with existing innovation diffusion literature.
- 5) Nonetheless, we believe our model is general enough to be extended to consider a variety of social diffusion phenomena, which we explore in the revised “Discussions” section. For instance, our model can be extended to cover innovation diffusion by providing a payoff advantage to one of the two strategies in the coordination game. If strategy $x = 1$ is the innovation, then the first term of the payoff function in Eq. (2a) would simply read $\sum_{w \in \mathcal{V} \setminus v} (1 + \alpha) x_w(t)$ where $\alpha > 0$ is the payoff advantage of the innovation with respect to the status quo.

In the revised Introduction, we have provided a more nuanced view of what our work (both the experiments and the model) focus on, viz. *social conventions*. In the Discussions, we then discuss some potential extension studies that could enable our work to connect with other diffusion phenomena.

- Ln 344. You cannot claim that these mechanisms are unique to humans. There is no evidence supporting this claim.

We agree with the Reviewer, and apologise for the wording choice. We have revised the manuscript, which now states that these mechanisms are “characteristic features of human decision-making processes”. In the revised manuscript, this is now found in line 419.

- I would like the authors to discuss whether the model could capture the fact that once a trend becomes mainstream it is not a trend anymore. Their model seems to capture only a binary set of options and a one diffusion process from status quo to innovation. This seems at odds with the dynamic evolution of norms and tastes in society. Discussing this point may help frame the paper in a larger narrative.

We thank the Reviewer for the interesting question.

- 1) In terms of the dynamic norm literature [24], [25], existing research does not suggest that the desire of individuals to seek trends stops once the trend hits the mainstream [26]. The effect of dynamic norms depends only on the *direction of change*, and not on the current majority option adopted by the population.
- 2) This fact is captured by our model, where the trend-seeking term provides a payoff advantage for the strategy that increased in popularity over the previous time-step, and does not consider whether the strategy is in the majority or minority. Nonetheless, one can make the trend-seeking term more realistic, by reducing the impact of trend-seeking for the strategy that is currently in the mainstream. This may be achieved by introducing a nonlinearity that decreases the magnitude of the trend-seeking term as $\frac{1}{n} \sum_{w \in \mathcal{V} \setminus \{v\}} x_w(t)$ tends to 1 or 0. Such a nonlinearity would slow down

the explosiveness of the diffusion process after the alternative becomes the majority strategy, which can be observed in some examples of real world diffusion, but would make the parametrisation significantly more complicated.

- 3) In the literature, many papers focusing on models for social diffusion make an assumption to simplify the problem and consider a binary choice between a status quo and an alternative, e.g. [5], [6], [11], [16]. In the real-world, there may be scenarios in which multiple alternative options are simultaneously introduced into a population; this can be captured by population game models in which players can choose among multiple options, see e.g. [27], [28]. Trend-seeking terms could be included in such models; we have shown the importance of trend-seeking in a binary decision-making model, and this gives motivation to explore the impact of dynamic norms in more complex models as future work.

We provided some limited discussion of points 2 and 3 on pg. 14 and 28 of the revised manuscript. Due to concerns around word limits for the manuscript, we are unfortunately unable to provide a more detailed exploration of these points.

- Ln 419 Your criteria for ‘irregular participants’ need attention. While criteria 1 and 3 make sense, criterion 2 is dangerous for the interpretability of the data. Why are stubborn people excluded when stubborn bots are such a crucial part of your paradigm? I would, at the very least, show in SI how the data is affected by the presence of stubborn people. Ln 423 You cannot exclude stubborn people (especially as defined as arbitrarily as “Play 0 for 2 rounds in a row when other players play 1”) because they are

We thank the Reviewer for the query. There are several points we wish to clarify.

- 1) There were a total of 3 stubborn individuals among the 180 registered participants, and so such individuals are rare. From these 3 trials, we could not derive any significant observations as to why they did not change their strategy at all, apart from some anecdotal remarks. As mentioned in the manuscript, we removed 4 trials due to the large number of irregular individuals. Among the remaining 16 trials, 2 trials had stubborn individuals; our description of how we treated these stubborn individuals during the parametrisation process appears to have generated some confusion, which we hope to clarify in the next point.
- 2) We have heavily revised the description of the parametrisation process in the “Methods” section of the paper, see pg. 35 and 39–40, to clarify how we excluded stubborn participants. Because the parameters of stubborn individuals are clear within our model, being $k_v = 1$ and $x_v(0) = 0$, we removed their switching rate data as there was no need to identify their parameters (β, k_v, r_v, b_v) from the switching rate data (see pg. 35). However, we *did not* remove the presence of stubborn individuals from the simulation scenarios recreating each experimental trial (see pg. 39–40); stubborn individuals still remained in the simulations during the parametrisation process and had an impact on the group dynamics and the strategy selections of regular players. However, we coded them deterministically as their responses were predetermined ($x_v(t) = 0$ for $t = 0, 1, \dots, 24$).
- 3) We conducted a robustness check where we included all 20 trials, as reported in SI Table S3. No significant differences in the parameter values were observed, and subsequent simulations with the new parameter values did not yield any conclusions that were qualitatively different (see SI Fig. S4).
- 4) When defining the rules to classify individuals as “stubborn,” we decided to adopt an extremely conservative definition, to avoid misclassification of regular players. Note that a necessary condition for a player to be stubborn is that the player selected the status quo strategy in all 24 rounds. However, such a condition is only necessary, but not sufficient (since an individual may be playing

the status quo strategy in all 24 rounds because a majority of the group were playing the status quo strategy in all 24 rounds). Hence, our criteria to be considered a stubborn individual required an individual to select the status quo strategy in all 24 rounds, *and then additionally* required that the individual *consistently* (that is, at least two rounds in a row) played the status quo when all the other players in the group played the alternative in these two rounds. In the context of the game, one would expect a non-stubborn individual faced with all others selecting the alternative for several repeated rounds to switch to the alternative in order to collect the reward (and in our model, $k_v < 1$).

*Minor Concerns*

- - The authors mention one of Damon Centola’s papers (on tipping points) but not others that are also
very relevant for their paper, namely the papers on complex contagion (Centola 2007, Guilbeault et al
2018), as well as the recent book by Damon (How behaviour spreads?). I would like to hear how this
paper relates to this literature.

We thank the Reviewer for the insightful suggestion. In our revised manuscript, we have included the
suggested references (in addition to a few others) and discussed the relation of our work to the complex
contagion literature. Our discussions are summarised into the following points:

- 1) Our work falls squarely into what is considered “complex contagion”, because the transmission of
the contagion (alternative) depends on a multitude of factors including how many other individuals
have the contagion. This contrasts “simple contagion” such as an infectious disease, where a single
infected individual is sufficient to pass on the contagion.
- 2) Our model and framework is applicable to “contagion” in which decisions can be revised repeat-
edly, e.g. using a handshake to greet someone. It is not applicable for “contagion” that cannot be
removed once it has “stuck” to the individual, e.g. installing solar panels on your house.
- 3) Our paper primarily seeks to advance the theoretical contributions to complex contagions, informed
by empirical data that we also gathered. The review chapter [14] identifies that the theoretical
contributions to complex contagion literature can be grouped into three areas: i) impact of network
topology for diffusion of complex contagion, ii) individual-level mechanisms governing diffusion,
and iii) design of network interventions and seeding strategies. Clearly, our contribution targets
the second aspect, as we are highlighting the impact of individual-level mechanisms of inertia and
trend-seeking.

- - Equation 4b, How is the value 0.4 selected? Please expand or cite literature to support your choice.

We thank the Reviewer for this comment, which has allowed us to improve our manuscript. In this
work, we are interested in quantifying the time taken to reach a “tipping point” (which can be delayed),
and subsequently the duration of the transition from the tipping point to a pervasive adoption of the
alternative. These two aspects are captured by \bar{T} and ΔT , respectively, and the nature of the diffusion
process is then indicated by the pair $(\bar{T}, \Delta T)$. We defined as “tipping point,” the point after which the
fraction of adopters of the alternative never decreases below 0.4.

To motivate and support our choice of setting such a threshold at 0.4, we have reported in the
Supplementary Information the new Fig. S8, which shows the velocity of the diffusion process (that
is, the rate of change), as a function of the current number of adopters of the alternative. Such a
plot positions the “tipping point” of the diffusion process approximately about 0.4. In fact, below the

threshold value of 0.4, the system mostly fluctuates about the meta-stable status-quo majority state (with
 the average velocity reaching a minimum); once the threshold is reached, the velocity of the diffusion
 process starts to grow.

 We have also conducted a robustness check by choosing different values for the threshold rather than
 0.4. In particular, the revised Supplementary Information, Figs. S9–S11 now show simulations where
 the threshold is set to 0.35 and 0.45, for 0.17, 0.25 and 0.33 fraction of committed minority. One can
 see that the key qualitative aspects of the diffusion patterns are unchanged, as described by ΔT and
 \bar{T} . This means that the features of explosive diffusion, and a delay is that dependent on the fraction of
 committed minority and fraction of explorers, are robust to the threshold value choice.

- SI Ln 56-63 I find this part very confusing. I don't see why missing trials means people are playing
 irrationally or at random.

 We have revised the manuscript to clarify this issue, through the following steps.

- 1) We have now removed speculation about what participants who missed multiple rounds are doing,
 i.e., whether they are playing randomly or irrationally. We still report some of their feedback
 as to what caused them to miss rounds, e.g., being distracted by the dog. This is in the revised
 Supplementary Information, Remark 2 on pg. 6
- 2) We now simply state that the presence of participants who missed multiple rounds, dropouts, and
 stubborn individuals created challenges for parametrisation of the others in the same trial, as the
 group dynamics may be heavily impacted by the irregular behaviour, and we cannot confidently
 characterise this irregular behaviour. So, we removed from the parametrisation data the 4 trials
 with multiple irregular individuals. This is in the revised manuscript, "Methods" section, pg. 35.
- 3) To test for robustness, we ran our parameterisation process including these 4 trials, and repeated our
 simulation study. The parameters obtained and the simulation results were robust to the inclusion
 of the 4 trials with multiple irregular participants. Note that this robustness study was included in
 the original Supplementary Information, and in the revised Supplementary Information is located
 in Section S3, Table S3 and Fig. S3 and S4.

- Fig S8, panel labels are wrong in the caption

 We thank the Reviewer for pointing out the typos. In our revised version of the Supplementary
 Information, we have updated the figure correcting the labels. Note that the comparison with the SIS
 model was expanded, following the suggestions of Reviewer 2.

- Please better define innovator and explorer. I thought you were using them interchangeably before I
 realized the two are different things.

 Thank you for pointing out this important observation, which made us realise that this distinction
 was not self-evident for the reader in the original manuscript. We have revised the manuscript and
 renamed "innovator" to "committed minority", leaving "explorers" unchanged. We hope this change in
 terminology helps to distinguish the two.

 In order to ensure this distinction is clear, we have also revised the Introduction so that the first paragraph
 describes how changes in social convention can come about. Here, we immediately introduce the concept
 of a committed minority, as the group that stubbornly push a new alternative. We then mention that

Fig. R1: Heatmap showing the answers given by the regular participants in the 20 trials. Rounds in which a participant selected 0 are coloured in green, while those in which he or she selected 1 are coloured in black. White boxes represent missed answers.

*explorers* may be the first to try out this alternative and may aid in unlocking diffusion.

We provide further reminders of the distinction at key points in the manuscript, including the description
of the experiments, and the description of the model setup (see e.g. pg. 8, 10 and 16.)

- Ln 122-126: Figure 1d should probably be complemented with a scatter plot or heatmap showing not
just the switching rate γ but also when the switch was likely to occur over the 24 rounds. I think this
visualization may help you also understand other dynamics occurring in your data that are obscured
by a simple histogram.

In Fig. R1 in this response letter, we report a heatmap, in which we show the temporal pattern of
responses during the experiment. Even though the readability of the figure may be hindered by the
different duration of the trials, it may help visualise the important role played by coordination, as well
as consistency in the participants' answers (probably a consequence of the presence of inertia in the
players' decision-making processes). Based on these observations, we performed a Wald–Wolfowitz
runs test to provide statistical significance to our intuition (as explained in the revised version of the
paper). However, from the figure, we do not visualise any other clear dynamics (besides the tendencies
to coordinate with others and to be consistent with the previous answers). For this reason, and for its
non-immediate readability, we have preferred to omit the figure from the main paper.

Fig. R2: Sample simulations with $n = 200$ agents for the model on (a–b) a Watts–Strogatz small-world network with average degree 20 and rewiring probability equal to (a) 0.5 and (b) 0.1; and (c–e) a Albert–Barabási scale-free network with average degree 20. In (a–c), the committed minority is positioned at random in the network, in (d) it is positioned in the hubs, and in (e) in the low degree nodes.

- Ln 337 I would suggest to the authors to speculate on how they predict different network structures
 (in particular the most common observed in social networks, e.g power law, sparsely connected com-
 munities, etc) may affect their results.

 We conducted preliminary simulations for our model on several classical network structures (that is,
 where the coordination term in the payoff function for an individual depends only on the fraction of
 adopters of the status quo and the alternative among the neighbours of that individual), including the
 Watts–Strogatz small-world network, and the Albert–Barabási scale-free network. The simulations are
 presented in Fig. R2 in the response letter.

 From these simulations, we see that the network structure can have a non-trivial impact on the diffusion
 dynamics, which is not surprising. For instance, it seems that the presence of clusters and few long range
 interactions (that is, comparing Fig. R2b to Fig. R2a) may produce a diffusion pattern that immediately
 reaches the tipping point, but whose diffusion is not explosive. On the other hand, the positioning of
 the committed minority seems also to play a key role. By comparing Figs. R2c–R2e, we observe very
 different diffusion patterns when the committed minority is positioned into (c) nodes at random, (d)
 hubs, and (e) low-degree nodes.

 The impact of the network structure can be highly complex, and requires careful and rigorous analysis.
 There are many complicating factors. For instance, in small-world networks, the diffusion dynamics

may depend heavily on the structure of the clusters, whether the committed minority are dispersed
 across many clusters or concentrated in specific clusters, and the presence of long-range interactions.
 In scale-free networks, whether the committed minority or explorers are at the high degree nodes may
 significantly change the dynamics. We believe that further speculation would not materially improve
 the manuscript, given that the limited space available would not allow for a clear discussion of the
 complexities arising from introducing a network structure. Rather, investigation into the effects of
 network structures for our proposed model deserves its own paper as a future work.

- - I wonder how the presence of dropout bots influenced your estimation of inertia. If these bots adopted
 whatever majority decision was observed in that trial (Ln 389), then the more dropouts you have the
 more the population will show inertia (resistance to change).

We thank the Reviewer for the insightful comment, which we clarify with the following points.

- 1) The presence of dropout bots did not influence our estimation of inertia because we parameterised
 the agent-based model using switching rates at the individual level, and the switching rate data
 of the dropout bots were not used.
- 2) Inertia relates to an individual's propensity to stick with *his/her own decision*. The dropout bots
 contributing to the majority decision would actually be reflected in the coordination mechanism,
 with parameter b_v . For instance, when more than 50% of the group (including committed minority
 bots) are selecting the *alternative*, the presence of dropout bots would contribute to promoting the
 diffusion, by encouraging any human participants currently selecting the status quo to switch to
 the alternative.
- 3) Nonetheless, the Reviewer makes a valid point that having too many dropout bots may potentially
 impact that overall group dynamics in a non-negligible manner that may be difficult to characterise.
 This is why we removed 4 trials from the parametrisation process, as these 4 trials had too many
 dropout bots and players that were missing several rounds (more than 33% of players displaying
 these irregular behaviours). We then checked the robustness of the parameter values obtained, by
 running the parametrisation process again with the inclusion of the 4 trials. These findings were
 reported in the Supplementary Information, in Table. S3.

We provided a brief remark on this in the Supplementary Information, in Remark 2.

- - Ln 466 Please describe in plain English the meaning of sw_v and its significance for measure you are
 calculating in Eq 6.

 We have revised the manuscript as suggested and provided an explanation in plain English concerning
 sw_v , see pg. 38 Ln 588. Roughly speaking, sw_v captures the total number of switches in the strategy
 individual v made before a full consensus was reached/the game finished. Then, sw_v/T^* is equivalent
 to the switching rate y_v as defined in Eq. (4d). Thus, a participant with higher sw_v switches more
 often between the two strategies during the game, which suggests explorer-like behaviour. A large sw_v
 provides a large positive contribution to the right hand side of Eq. (6), and makes it more likely for
 that participant to be identified as an explorer.

REFERENCES

- [1] D. Centola, J. Becker, D. Brackbill, and A. Baronchelli, "Experimental evidence for tipping points in social convention,"
 *Science*, vol. 360, no. 6393, pp. 1116–1119, 2018.
- [2] S. Moscovici, E. Lage, and M. Naffrechoux, "Influence of a consistent minority on the responses of a majority in a color
 perception task," *Sociometry*, pp. 365–380, 1969.

- [3] W. Wood, S. Lundgren, J. A. Ouellette, S. Busceme, and T. Blackstone, “Minority influence: a meta-analytic review of social
influence processes,” *Psychological Bulletin*, vol. 115, no. 3, p. 323, 1994.
- [4] M. Muthukrishna and M. Schaller, “Are collectivistic cultures more prone to rapid transformation? computational models of
cross-cultural differences, social network structure, dynamic social influence, and cultural change,” *Personality and Social
Psychology Review*, vol. 24, no. 2, pp. 103–120, 2020.
- [5] J. Henrich, “Cultural transmission and the diffusion of innovations: Adoption dynamics indicate that biased cultural transmission
is the predominate force in behavioral change,” *American Anthropologist*, vol. 103, no. 4, pp. 992–1013, 2001.
- [6] H. Peyton Young, “Innovation Diffusion in Heterogeneous Populations: Contagion, Social Influence, and Social Learning,”
*American Economic Review*, vol. 99, no. 5, pp. 1899–1924, 2009.
- [7] C. Bicchieri, *Philosophy of social science: A new introduction*. Oxford University Press Oxford, 2014, ch. Norms, Conventions,
and the Power of Expectations, pp. 208–229.
- [8] D. Lewis, *Convention: A Philosophical Study*, 1st ed. Hoboken NJ, US: Wiley-Blackwell, 2002.
- [9] A. Marmor, *Social Conventions: From Language to Law*, 1st ed. Princeton NJ, US: Princeton University Press, 2009.
- [10] H. Peyton Young, “The Evolution of Conventions,” *Econometrica: Journal of the Econometric Society*, pp. 57–84, 1993.
- [11] A. Montanari and A. Saberi, “The spread of innovations in social networks,” *Proceedings of the National Academy of Sciences*,
961 vol. 107, no. 47, pp. 20 196–20 201, 2010.
- [12] G. E. Kreindler and H. P. Young, “Rapid innovation diffusion in social networks,” *Proceedings of the National Academy of
Sciences*, vol. 111, no. 3, pp. 10 881–10 888, 2014.
- [13] S. De, D. S. Nau, X. Pan, and M. J. Gelfand, “Tipping points for norm change in human cultures,” in *International Conference
on Social Computing, Behavioral-Cultural Modeling and Prediction and Behavior Representation in Modeling and Simulation*.
Springer, 2018, pp. 61–69.
- [14] D. Guilbeault, J. Becker, and D. Centola, *Complex Contagions: A Decade in Review*. Springer, 2018, pp. 3–25.
- [15] J. Maynard Smith, *Evolution and the Theory of Games*, 1st ed. Cambridge, UK: Cambridge University Press, 1982.
- [16] H. Peyton Young, “The Evolution of Social Norms,” *Annual Review of Economics*, vol. 7, no. 1, pp. 359–387, 2015.
- [17] C. T. Bauch and D. J. D. Earn, “Vaccination and the theory of games,” *Proc. Natl. Acad. Sci. US*, vol. 101, no. 36, pp.
971 13 391–13 394, 2004.
- [18] E. Rogers, *Diffusion of Innovations*, 5th ed. New York NY, US: Free Press, 2003.
- [19] S. Asadi, C. D. Cappa, S. Barreda, A. S. Wexler, N. M. Bouvier, and W. D. Ristenpart, “Efficacy of masks and face coverings
in controlling outward aerosol particle emission from expiratory activities,” *Scientific Reports*, vol. 10, no. 1, Sep. 2020.
- [20] E. P. Fischer, M. C. Fischer, D. Grass, I. Henrion, W. S. Warren, and E. Westman, “Low-cost measurement of face mask
efficacy for filtering expelled droplets during speech,” *Science Advances*, vol. 6, no. 36, p. eabd3083, Aug. 2020.
- [21] R. Amato, L. Lacasa, A. Díaz-Guilera, and A. Baronchelli, “The dynamics of norm change in the cultural evolution of
language,” *Proceedings of the National Academy of Sciences*, vol. 115, no. 33, pp. 8260–8265, 2018.
- [22] A. Baronchelli, “The emergence of consensus: a primer,” *Royal Society open science*, vol. 5, no. 2, p. 172189, 2018.
- [23] J. Xie, S. Sreenivasan, G. Korniss, W. Zhang, C. Lim, and B. K. Szymanski, “Social consensus through the influence of
committed minorities,” *Physical Review E*, vol. 84, p. 011130, Jul 2011.
- [24] G. Sparkman and G. M. Walton, “Dynamic Norms Promote Sustainable Behavior, Even If It Is Counternormative,”
*Psychological Science*, vol. 28, no. 11, pp. 1663–1674, 2017.
- [25] C. R. Mortensen, R. Neel, R. B. Cialdini, C. M. Jaeger, R. P. Jacobson, and M. M. Ringel, “Trending norms: A lever for
encouraging behaviors performed by the minority,” *Social Psychological and Personality Science*, vol. 10, no. 2, pp. 201–210,
2019.
- [26] D. D. Loschelder, H. Siepelmeyer, D. Fischer, and J. A. Rubel, “Dynamic norms drive sustainable consumption: Norm-based
nudging helps café customers to avoid disposable to-go-cups,” *Journal of Economic Psychology*, vol. 75, p. 102146, 2019.
- [27] J. Hofbauer, K. Sigmund *et al.*, *Evolutionary games and population dynamics*. Cambridge university press, 1998.
- [28] W. H. Sandholm, *Population games and evolutionary dynamics*. MIT press, 2010.

Reviewers' Comments:

Reviewer #1:

Remarks to the Author:

I appreciate the work done on the revision. While some of the issues have been resolved, I still have a number of reservations listed below.

1. There is still a lack of direct evidence showing that the social diffusion pattern in the experiment was due to individual inertia and/or trend-seeking. The discussion was based on reasoning from results on the population level, not direct measurement on the individual level (e.g., the discussion on Page 11). I would suggest the authors look more closely into the individual level decision making processes in the experiment and examine the role of inertia and trend-seeking directly. Otherwise, it's not clear whether this ABM is directly reflecting the mechanism behind the empirical data.

2. Related to the above, what's the key difference between the current experiment and the experiment in Centola et al. (2018)? Can the ABM also explain the behavioral patterns in Centola et al. or similar work? Would the authors expect their ABM to explain the empirical data better than the model of Centola et al.?

3. Figure S8 shows the rationale for choosing 0.4 as the tipping point. The robustness tests through Figures S9-11 are helpful. However, I would suggest the authors give a theoretical definition of "tipping point," since this is a very important concept for the paper. On Page 3 you note that "A tipping point is reached if the combination of committed minority and explorers adopting the alternative reach a critical mass, which then sparks a rapid diffusion to the rest of the population, who we term non-explorers." This seems to suggest that tipping point is a time point in the experiment/simulation process whenever the diffusion starts to explode. However, you also operationalized it as the time that the alternative reaches 0.4. Though practically these may mean similar time points, I'm not sure whether it's appropriate to equate the two definitions, because technically the explosiveness may happen at different proportions depending on model assumptions (e.g., Andreoni et al., 2021). It seems that what the authors used is not a "tipping point" but a milestone that marks that the diffusion is substantial. Maybe you can consider replacing the term. On the other hand, if you really want to use tipping point, then the concept of "transition time" becomes problematic because given that the explosiveness has happened, the transition time should be short anyway. I think the concepts of "tipping point" and "transition time" may not be compatible and I hope the authors can clarify it.

Andreoni, J., Nikiforakis, N., & Siegenthaler, S. (2021). Predicting social tipping and norm change in controlled experiments. *Proceedings of the National Academy of Sciences*, 118(16).

Reviewer #2:

Remarks to the Author:

Dear Authors,

Thank you for a comprehensive consideration of mine and the other reviewers' comments. I think you have made good use of the suggestions provided. Your clarification of context in the introductory section gives the right context for the remainder of the work. You've addressed my concerns re: the bass model well, and the figures communicate much more clearly.

I'm happy to recommend your revision for publication.

Best of luck in your future work,

Reviewer #3:

Remarks to the Author:

The authors have satisfactorily addressed all my concerns. I appreciate the revision done by the authors, which showed their thorough work. Highlighting the edited parts of the original manuscript was a really welcome touch.

There are still a few minor aspects that the authors might want to tackle before the paper is accepted.

I agree that using the term "committed minority" is less ambiguous than using the previous term "innovators". However, Figure 5 x-label still uses the term "innovators"

Similarly, the main text and Figure 5 use $|I|/n$ and $|C|/n$ interchangeably to describe the proportion of committed minority.

Good luck,

Niccolo Pescetelli

Collective patterns of social diffusion are shaped by individual
inertia and trend-seeking
NCOMMS-20-37555A – Response to Reviewers

REVIEWER #1

I appreciate the work done on the revision. While some of the issues have been resolved, I still have a
number of reservations listed below.

We thank the Reviewer for the care in reviewing the revised manuscript and we are pleased to that you
appreciate the revision carried out. We are also thank you for these additional comments, which we have
carefully considered as part of revision process. In the following, we provide a point-to-point response
to the your comments, and describe adjustments to the manuscript (main article and methods) and the
supplementary information document.

- 1) There is still a lack of direct evidence showing that the social diffusion pattern in the experiment was
due to individual inertia and/or trend-seeking. The discussion was based on reasoning from results on
the population level, not direct measurement on the individual level (e.g., the discussion on Page 11).
I would suggest the authors look more closely into the individual level decision making processes in
the experiment and examine the role of inertia and trend-seeking directly. Otherwise, it's not clear
whether this ABM is directly reflecting the mechanism behind the empirical data.

We thank the Reviewer for this comment, which has allowed us to provide better motivation for the
presence of inertia and trend-seeking in the agent-based model.

We have performed new analysis to further strengthen our arguments, as suggested by the Reviewer. In
particular, we conducted a regression analysis with a fixed effects estimator on the experimental data, at
the *individual level*; this new analysis and the results are described on pg. 11 of the revised manuscript
(blue text), with the full details reported in the revised Supplementary Information, Section 2. The
results of the analysis suggest that both inertia and trend-seeking have a statistically significant role
during each individual's decision-making process, in the experimental data.

We then re-performed the trend-seeking analysis on Page 11 to focus on the individual-level behaviour
(while the previous analysis was conducted at the population level). We focused on what *an individual*
would do depending on the trend, in particular in a 50%-50% scenario (in which social coordination
does not favour any of the choices). The analysis continues to suggest that trend-seeking plays an
important role in individual behaviour. We have reported the outcome of the analysis on Page 11 of
the revised manuscript (with full details in the revised Supplementary Information, Section S2). Note
that the existing analysis on the presence of inertia on pg. 11 was already at the individual level.

The additional individual-level analyses, including the regression analysis, provide strong and direct
evidence on the presence and importance of inertia and trend-seeking within our empirical data, and
together with the paramtrisation process of the ABM, justify inertia and trend-seeking being presence
in the ABM.

2) Related to the above, what's the key difference between the current experiment and the experiment in
 Centola et al. (2018)? Can the ABM also explain the behavioral patterns in Centola et al. or similar
 work? Would the authors expect their ABM to explain the empirical data better than the model of
 Centola et al.?

We thank the Reviewer for this comment. While Centola *et al.* (2018) served as the primary inspiration
 for our work, including the focus on social conventions, there are several key differences. We answer
 each of the three questions posed by the Reviewer in the order they were posed.

The fundamental concept of our experiment and that of Centola *et al.* is the same: participants are
 encouraged to coordinate (reflected in their reward). This aims to reflect how social conventions work
 in the real-world [1], [2]. Regarding the key differences:

- a) The main difference is in the interaction mechanism and the information gained from interacting.
 In our implementation, every individual updates the choice at the same time, and can observe
 *group-level information*, viz. the fraction the group selecting the two products, *Eta* and *Tao*.
 From a network perspective, this is equivalent to an all-to-all interaction network. In contrast, in
 Centola's experiment, in each round two participants were paired together (pairwise interactions
 drawn uniformly at random in each round). Thus, at each interaction, a participant only learned
 of information about a single participant among the group. We have included a statement on this
 difference in the revised manuscript, in the Methods (see pg. 31). In real-world social conventions,
 pairwise interactions may occur for greetings (handshakes vs bowing) while group interactions
 may occur when people decide to walk on the left or right side of the footpath.
- b) A second, less important difference, was the number of rounds played. In our implementation,
 we restricted the game to 24 total rounds for practical purposes. Centola ensured each participant
 was involved in a minimum number of interactions, but the length was also dependent on how
 many rounds it took to finish the equivalent of our Stage I).

Regarding whether our ABM can also explain the behavioural patterns in Centola *et al.* or similar
 work:

- a) Given that the interaction structure is different (see above), our ABM cannot be directly applied
 to Centola's paradigm.
- b) It might be possible to extend our model to reflect a more general interaction structure than
 all-to-all, by modifying the first term in (2a) and (2b) for coordination and also the third term on
 trend-seeking. We do not believe this should be pursued as part of this manuscript. This is because
 significant new material would be required to present a rigorous treatment on i) describing the
 modifications, such as to capture pairwise interactions, ii) running new experiments/getting data
 based on pairwise interactions to parametrise the modified model (since our parameters b_v, k_v, r_v
 and β were secured using our experimental data based on group interactions), and iii) reporting
 and discussing all the new results. This amount of material would best be presented as a separate
 manuscript rather than additions to our already comprehensive study, and would be suitable as
 future work; our work focuses on the impact of the mechanisms of inertia and trend-seeking
 during social diffusion.

Regarding the question of whether we would expect our ABM to explain the empirical data better
 than the model of Centola et. al. (2018):

- a) We want to stress that our model should not be seen as a replacement for, or improvement on,
 Centola's model and experiments, given the different experimental setups regarding the interaction
 structures. Our model aims to capture different potential mechanisms (inertia and trend-seeking)

and explore their consequences on diffusion of new social conventions.

b) Our ABM does explain *our* empirical data better than an ABM based only on coordination (see Fig. 3 and the associated results in the main manuscript).

c) While generalisations of our model to different interaction structures and further exploration against other data sets are out of the scope of the current paper, we have added a sentence in the discussion to highlight its importance in the future research.

- 3) Figure S8 shows the rationale for choosing 0.4 as the tipping point. The robustness tests through Figures S9-11 are helpful. However, I would suggest the authors give a theoretical definition of “tipping point,” since this is a very important concept for the paper. On Page 3 you note that “A tipping point is reached if the combination of committed minority and explorers adopting the alternative reach a critical mass, which then sparks a rapid diffusion to the rest of the population, who we term non-explorers.” This seems to suggest that tipping point is a time point in the experiment/simulation process whenever the diffusion starts to explode. However, you also operationalized it as the time that the alternative reaches 0.4. Though practically these may mean similar time points, I’m not sure whether it’s appropriate to equate the two definitions, because technically the explosiveness may happen at different proportions depending on model assumptions (e.g., Andreoni et al., 2021). It seems that what the authors used is not a “tipping point” but a milestone that marks that the diffusion is substantial. Maybe you can consider replacing the term. On the other hand, if you really want to use tipping point, then the concept of “transition time” becomes problematic because given that the explosiveness has happened, the transition time should be short anyway. I think the concepts of “tipping point” and “transition time” may not be compatible and I hope the authors can clarify it.

Andreoni, J., Nikiforakis, N., & Siegenthaler, S. (2021). Predicting social tipping and norm change in controlled experiments. *Proceedings of the National Academy of Sciences*, 118(16).

We thank the Reviewer for providing a comprehensive argument around the concerns with our use of the term “tipping point”. We agree with the Reviewer that our use of the term “tipping point” could be confusing, in the way it was operationalized in the previous version of the manuscript.

For this reason, we have followed your suggestion. We have removed the term “tipping point” throughout the manuscript and Supplementary Information, whenever referring to the variable \bar{T} and associated discussions. We have opted to refer to \bar{T} explicitly and exclusively as the take-off time, and we describe \bar{T} as identifying the time at which the diffusion process reaches a milestone, after which it takes off. We can consider \bar{T} as the point at which diffusion takes off because i) a substantial number (but a non-majority) of the agents have selected the alternative, ii) the diffusion process is exiting (or has exited) the meta-stable status quo majority, and iii) the velocity plots in Fig. S8 in the Supplementary Information indicate a positive acceleration of the diffusion process. We hope this clarification will allow to avoid any possible confounding with the literature on the tipping point.

We would also like to thank you for providing the interesting and very recent reference. We have incorporated the reference as part of the literature on social diffusion in the Introduction, and separately in the Discussions section.

REVIEWER #2

Dear Authors,

Thank you for a comprehensive consideration of mine and the other reviewers' comments. I think you
 have made good use of the suggestions provided. Your clarification of context in the introductory section
 gives the right context for the remainder of the work. You've addressed my concerns re: the bass model
 well, and the figures communicate much more clearly.

I'm happy to recommend your revision for publication.

Best of luck in your future work,

Thank you for your kind words and for the positive recommendation. We also want to thank you again
 for the insightful comments you gave us in the previous round of the review process that allowed us to
 improve the quality of the manuscript.

REVIEWER #3

The authors have satisfactorily addressed all my concerns. I appreciate the revision done by the authors,
 which showed their thorough work. Highlighting the edited parts of the original manuscript was a really
 welcome touch.

There are still a few minor aspects that the authors might want to tackle before the paper is accepted.

I agree that using the term "committed minority" is less ambiguous than using the previous term "inno-
 vators". However, Figure 5 x-label still uses the term "innovators"

Similarly, the main text and Figure 5 use $|I|/n$ and $|C|/n$ interchangeably to describe the proportion of
 committed minority.

Good luck, Niccolo Pescetelli

Dear Niccolo Pescetelli,

We are pleased to read that you are satisfied by how we have addressed your concerns on the previous
 version of the manuscript, and that you appreciate our revision. We also thank you for having pointed out
 the typos with Figure 5, which we have amended in our revised version of the manuscript.

REFERENCES

- [1] C. Bicchieri, *Philosophy of social science: A new introduction*. Oxford University Press Oxford, 2014, ch. Norms, Conventions,
 and the Power of Expectations, pp. 208–229.
 [2] D. Lewis, *Convention: A Philosophical Study*, 1st ed. Hoboken NJ, US: Wiley-Blackwell, 2002.

Reviewers' Comments:

Reviewer #1:

Remarks to the Author:

Thank you for your responsiveness. I have only a few final comments.

1) Section 2 in the supplementary information is very helpful. It has addressed the individual decision-making processes well. Only one more minor comment on the regression on Page 9 in SI: There might be an omitted variable—the absolute proportion of the choices. Participants have the tendency to conform with the majority. Thus, there may be a correlation between the major choice and the group trend. People may argue that the participants are not following the trend, but just following the majority. Including this omitted variable should not change the conclusion of the regression because in S2.3, the authors showed that the group trend is making a difference even when the proportions are 50%-50%. However, just for the completeness of the regression, ideally the authors could include the proportions into the regression and show that group trend predicts the behavior even beyond the absolute proportions. Other than that, the authors have addressed my question.

2) I appreciate the authors' explanation. I have no further questions on this.

3) Since the authors have removed the term "tipping point," I have no further questions.

Collective patterns of social diffusion are shaped by individual
inertia and trend-seeking
NCOMMS-20-37555B – Response to Reviewers

Mengbin Ye, Lorenzo Zino, Žan Mlakar, Jan Willem Bolderdijk, Hans Risselada, Bob M. Fennis,
Ming Cao

REVIEWER #1

Thank you for your responsiveness. I have only a few final comments.

We would like to thank the Reviewer for the insightful comments provided in the previous round of the
review process and we are happy to read our responses were well received. We would also like to thank
you for these final comments, which we carefully considered in preparing this revision.

1) Section 2 in the supplementary information is very helpful. It has addressed the individual decision-
making processes well. Only one more minor comment on the regression on Page 9 in SI: There
might be an omitted variable—the absolute proportion of the choices. Participants have the tendency
to conform with the majority. Thus, there may be a correlation between the major choice and the
group trend. People may argue that the participants are not following the trend, but just following the
majority. Including this omitted variable should not change the conclusion of the regression because
in S2.3, the authors showed that the group trend is making a difference even when the proportions
are 50%-50%. However, just for the completeness of the regression, ideally the authors could include
the proportions into the regression and show that group trend predicts the behavior even beyond the
absolute proportions. Other than that, the authors have addressed my question.

We are pleased to hear that you appreciated the new regression analysis (now in Supplementary Note 2).
We would also like to thank you for this further suggestion. In our revised version of the manuscript,
we have now included the fraction of participants choosing the alternative in the previous round as a
predictor. The result of this additional more comprehensive regression analysis allows us to conclude
that all the three predictors (fraction of participants choosing the alternative in the previous round,
the participant's choice in the previous round, and group trend) have a statistically significant impact
on the current choice. Our findings are consistent with your comment: including this new predictor
does not change the conclusion of the regression. We can thus confirm that the three mechanisms of
social coordination, inertia, and trend-seeking play an important role in the decision-making process
and provides further evidence toward their inclusion in the agent based model. The new regression
analysis and results have been incorporated into the revised manuscript, including the main article
(pg 11, and Table 1) and the Supplementary Information (Supplementary Note 2).

2) I appreciate the authors' explanation. I have no further questions on this.

We are glad to hear that you have appreciated our explanations.

3) Since the authors have removed the term “tipping point,” I have no further questions.

We thank you again for the comment provided in the previous round of review.